# 3D-Printed Hydrogel for Diverse Applications: A Review

**DOI:** 10.3390/gels9120960

**Published:** 2023-12-07

**Authors:** Arpana Agrawal, Chaudhery Mustansar Hussain

**Affiliations:** 1Department of Physics, Shri Neelkantheshwar Government Post-Graduate College, Khandwa 450001, India; agrawal.arpana01@gmail.com; 2Department of Chemistry and Environmental Science, New Jersey Institute of Technology, Newark, NJ 07102, USA

**Keywords:** hydrogels, 3D printing, biomedical applications, natural hydrogels, synthetic hydrogels

## Abstract

Hydrogels have emerged as a versatile and promising class of materials in the field of 3D printing, offering unique properties suitable for various applications. This review delves into the intersection of hydrogels and 3D printing, exploring current research, technological advancements, and future directions. It starts with an overview of hydrogel basics, including composition and properties, and details various hydrogel materials used in 3D printing. The review explores diverse 3D printing methods for hydrogels, discussing their advantages and limitations. It emphasizes the integration of 3D-printed hydrogels in biomedical engineering, showcasing its role in tissue engineering, regenerative medicine, and drug delivery. Beyond healthcare, it also examines their applications in the food, cosmetics, and electronics industries. Challenges like resolution limitations and scalability are addressed. The review predicts future trends in material development, printing techniques, and novel applications.

## 1. Introduction

The rapid advancement of 3D printing technology has revolutionized how we conceive, design, and produce objects across various industries. From aerospace to healthcare, 3D printing has ushered in an era of unparalleled customization, reduced lead times, and intricate geometries that were previously unattainable using traditional manufacturing methods [1]. By layering materials to construct 3D objects directly from digital designs, 3D printing has transcended conventional limitations, allowing for unprecedented levels of innovation. Applications of 3D printing are extensive and span a myriad of sectors and materials/composite materials [2]. In aerospace, the technology has enabled the production of complex lightweight components with enhanced performance characteristics. In the automotive industry, it has facilitated the rapid prototyping of parts and the creation of specialized tools on-demand. Similarly, healthcare has experienced transformative changes, with 3D printing employed to craft patient-specific implants, anatomical models for surgical planning, and even bioengineered tissues [3,4].

Amidst this technological evolution, one class of materials has emerged as particularly intriguing for 3D printing: hydrogels. Hydrogels are three-dimensional networks of hydrophilic polymers that possess the remarkable ability to absorb and retain large amounts of water or biological fluids [5]. This unique property imbues hydrogels with an inherent compatibility with living systems, making them an ideal candidate for a range of applications that intersect with the life sciences. Hydrogels can be of several types, including natural, synthetic, hybrid, and physically and chemically crosslinked hydrogels. Notably, hydrogels’high water content and tunable mechanical properties render them an exceptional material for 3D printing. Various 3D printing technologies can be successfully employed for the printing of hydrogels, including extrusion-based 3D printing [6], inkjet-based printing [7], stereolithography [8], laser-assisted bioprinting [9], digital light processing [10], etc. Naghieh et al. [11] explored the printability of 3D hydrogel scaffolds, focusing on the influence of hydrogel composition and printing parameters, and contributedto optimizing 3D printing processes for creating intricate structures used in tissue engineering and other applications. Kalyan and Kumar [12] provided insights into the broad spectrum of 3D printing applications, including tissue engineering, medical devices, and drug delivery, and served as a comprehensive overview of the current state of 3D printing technology in healthcare and biomedicine.

The capability to engineer the stiffness, porosity, and degradation rate of hydrogels precisely allows for tailoring constructs to match the specific requirements of the target application. This adaptability opens doors to advancements in tissue engineering, drug delivery, wound healing, and beyond. Moreover, the compatibility of hydrogels with cells and tissues offers the potential to create biomimetic structures that closely mimic natural physiological environments, thereby enhancing their utility in regenerative medicine and disease modeling. Figure 1 schematically shows the various types of hydrogels and their printing techniques that can be employed for several applications in various sectors. In a recent review, Kapusta and colleagues [13] delved into utilizing antimicrobial natural hydrogels in the realm of biomedicine, examining attributes, potential applications, and the challenges presented by hydrogels. This work sheds light on the manifold opportunities these hydrogels hold within various medical fields. Kaliaraj et al. [14] directed their attention to the auspicious uses of hydrogels in the realm of 3D printing technology, where the emphasis lies in highlighting the versatile nature of hydrogels as viable materials for 3D printing, underscoring their pivotal role in this innovative manufacturing technique. Zhang et al. [15] presented an exhaustive review that encapsulates the latest advancements in the domain of 3D printing concerning sturdy hydrogels. Their in-depth exploration provides valuable insights into the evolution of resilient hydrogel materials and their applications, thereby contributing significantly to the burgeoning field of 3D printing.

As we embark on this journey through the realm of hydrogels in 3D printing, this review article aims to explore their multifaceted potential, discussing the diverse range of hydrogel materials, printing techniques, and applications within which they play a pivotal role. By delving into the synergy between hydrogels and 3D printing, we seek to unravel the novel avenues for innovation that this convergence has unlocked, ultimately shaping the landscape of industries ranging from healthcare to consumer products.

## 2. Hydrogels: An Overview

Hydrogels are a class of materials with a three-dimensional network structure composed of hydrophilic polymers that can absorb and retain significant amounts of water or biological fluids. This unique property gives hydrogels a resemblance to natural soft tissues, making them valuable for a wide range of applications across various fields. This section will discuss the classification of hydrogels, their properties, and their applications.

### 2.1. Classification of Hydrogels

The choice of hydrogel material significantly influences the printability, biocompatibility, and mechanical properties of 3D-printed constructs. Researchers need to carefully consider these factors when selecting a hydrogel material for their specific application to ensure optimal performance and desired outcomes. It should be noted here that the hydrogels can be classified based on their composition and crosslinking methods. On the basis of composition, they can be natural hydrogels or synthetic hydrogels. 

#### 2.1.1. Natural Hydrogels

Natural hydrogels can be derived from natural polymers such as alginate, collagen, chitosan, and hyaluronic acid and often possess inherent bioactivity and biocompatibility.These hydrogels can be challenging to 3D print due to their complex rheological properties, including viscosity and shear-thinning behavior. Modifications, such as optimizing gelation kinetics or mixing with other materials, might be necessary to enhance printability. Also, they tend to be biocompatible due to their similarity to the extracellular matrix of tissues and hence promote cell adhesion, proliferation, and differentiation, making them valuable for applications involving direct interaction with living cells. Başyiğit et al. [16] explored soy-protein-based hydrogels enhanced with locust bean gum, investigating their mechanical properties and release characteristics. This research advances our comprehension of the potential utility of these hydrogels in applications such as controlled drug delivery systems. In other work, Xin and colleagues [17] developed a specialized okra-based hydrogel for chronic diabetic wounds, addressing a significant healthcare challenge and underscoring the promise of natural hydrogels in wound care. In the same year, Haghbin et al. [18] embarked on creating a Persian gum-based hydrogel loaded with gentamicin-loaded natural zeolite, conducting a comprehensive study of its properties both in vitro and in silico. This investigation carries implications for drug delivery and infection management, underscoring the adaptability of natural hydrogels. The biodeterioration of stone monuments and its correlation with cyanobacterial biofilm growth was also investigated [19]. They also explored the use of essential oils in natural hydrogels. This study has practical applications in preserving cultural heritage and the environmentally friendly management of biofilms. Huang et al. [20] directed their efforts toward the biofabrication of a natural Au/bacterial cellulose hydrogel for bone tissue regeneration via insitu fermentation. This research addresses the pressing need for biocompatible materials in regenerative medicine, underscoring the potential of natural hydrogels in the field of tissue engineering. 

The construction and properties of physically cross-linked hydrogels based on natural polymers werecarried out by Yang et al. [21]. Their work bears wide-ranging implications in the field of biomedicine, including applications in drug delivery, wound healing, and tissue engineering. Xu et al. [22] delved into examining chitosan-based high-strength supramolecular hydrogels intended for 3D bioprinting, which advances the bioprinting field and highlights the promise of natural hydrogels as bioink materials. Cai and colleagues [23] reported on the potential of Laponite^®^-incorporated oxidized alginate–gelatin composite hydrogels for use in extrusion-based 3D printing, which holds great significance within the realm of 3D printing technology, illustrating how natural hydrogels can be seamlessly integrated into advanced manufacturing processes. These collective studies contribute to the growing domain of natural hydrogels, showcasing their adaptability and potential across diverse applications, from healthcare to cultural preservation. It should be noted that the mechanical strength and stability of these hydrogels may be relatively lower, which can limit their use in load-bearing applications. However, they excel in creating biomimetic environments for cell growth. 

#### 2.1.2. Synthetic Hydrogels

Contrary to natural hydrogels, synthetic hydrogels can be obtained from synthetic polymers like polyethylene glycol (PEG), polyacrylamide, and polyvinyl alcohol (PVA), and hence, the properties of synthetic hydrogels can be precisely controlled through chemical synthesis. These hydrogels generally exhibit more consistent and predictable printing behavior, and their properties can be finely tuned to match the desired viscosity and flow characteristics for different printing techniques. The biocompatibility of such hydrogels depends on the specific polymer and crosslinking chemistry used. Modifications can be made to enhance biocompatibility, and biofunctionalization can be applied to improve cell interactions. Apart from this, they also offer superior control over mechanical properties, which can be engineered to mimic a wide range of tissue stiffness, making them suitable for applications where mechanical support is crucial. Van Velthoven et al. [24] conducted a study onthe entrapment of growth factors within synthetic hydrogels, specifically examining the bioactive bFGF-functionalized polyisocyanide hydrogels. Their goal was to enhance the controlled release and biological activity of growth factors, potentially benefiting biomedical applications. 

Saccone et al. [25] pioneered a novel additive manufacturing (AM) technique based on visible light photopolymerization, called Hydrogel Infusion Additive Manufacturing (HIAM), which allows for the creation of a wide range of micro-architected metals and alloys using a single photoresin composition. In this process, 3D-architected hydrogel scaffolds are employed as platforms for in situ material synthesis reactions, as depicted schematically in Figure 2a. The first step in producing metal microlattices involves using DLP to print architected organogels based on N,N-dimethylformamide (DMF), and polyethylene glycol diacrylate (PEGda). The DLP printing phase determines the final shape of the part, and following printing, a solvent exchange replaces DMF with water, transforming the organogels into hydrogels. These hydrogel structures are then soaked in a solution containing metal salt precursors, allowing the metal ions to infiltrate and expand the hydrogel scaffold. Calcination in an air environment converts the metal–salt-swollen hydrogels into metal oxides, and a subsequent reduction in a forming gas mixture (95% N_2_, 5% H_2_) yields metal or alloy replicas that match the initially designed architecture. Throughout this process, the shape of the part, defined during DLP printing, is retained, with each dimension experiencing approximately 60–70% linear shrinkage, accompanied by an approximate mass loss of 65–90% during calcination. To illustrate the versatility of HIAM when compared to previous visible light photopolymerization AM techniques, they employed HIAM to create octet lattice structures using various materials such as copper (as shown in process steps in Figure 2b–e), nickel, silver, their alloys, and more complex materials like the high-entropy alloy CuNiCoFe and the refractory alloy W-Ni (Figure 2f). Additionally, they demonstrated the fabrication of multi-material structures, such as Cu/Co (Figure 2g–h). HIAM stands out for its capacity for parallelization, allowing multiple organogels to be printed simultaneously, swelled in distinct solutions, and subsequently calcined/reduced collectively. In Figure 2i, one can observe the simultaneous calcination of eight hydrogel lattices (precursors for Cu, CuNi, CuNiCoFe, and CuNiCoFeCr), resulting in the formation of oxides. The scale bars shown in Figure 2b,c are 5 mm; Figure 2d–f 1 mm; Figure 2g 1 cm; Figure 2h 2 mm; and Figure 2i 2 cm. Figure 2j–m represents the SEM image of Cu and CuNi samples from an overhead view (Figure 2j,l) and a single junction node (Figure 2k,m), respectively, and revealed that the Cu and CuNi samples retained their octet lattice shape throughout the thermal treatment, with beam diameters approximately around 40 µm.

Adjuik et al. [26] presented an extensive review that delves into the degradability of both bio-based and synthetic hydrogels, with a particular emphasis on their potential as sustainable soil amendments. This comprehensive review explores the environmental implications and ecological consequences of using hydrogels in agriculture, shedding light on the sustainability and compatibility of these materials when applied to soil-enhancement practices. The 3D-printable synthetic hydrogel designed as an immobilization matrix for continuous synthesis involving fungal peroxygenases has also been reported [27]. This innovative approach holds promise in the field of biocatalysis, enabling efficient and uninterrupted production of valuable compounds. The synthetic hydrogel matrix offers a stable and controlled environment for enzyme activity, facilitating the development of sustainable and efficient biochemical processes. Yuk et al. [28] created a high-performance 3D-printable ink utilizing one of the most commonly used conducting polymers, poly(3,4-ethylenedioxythiophene):polystyrene sulfonate (PEDOT:PSS), to harness the capabilities of advanced 3D printing in producing conducting polymer structures. To ensure suitable rheological properties for 3D printing, they developed a paste-like conducting polymer ink, which is derived from cryogenically freezing an aqueous PEDOT:PSS solution Figure 3a, followed by lyophilization and controlled re-dispersion in a mixture of water and dimethyl sulfoxide (DMSO) (as shown in Figure 3b). The resulting conducting polymer ink demonstrates exceptional 3D-printing capabilities, enabling high-resolution printing (with a resolution of over 30 µm), the creation of structures with a high aspect ratio (exceeding 20 layers), and consistent fabrication of conducting polymers. These structures can also be seamlessly integrated with other 3D-printable materials, such as insulating elastomers, using multi-material 3D printing. By subjecting the 3D-printed conducting polymers to dry annealing, they achieved highly conductive microstructures that remain flexible in their dry state, which can be easily transformed into a soft, highly conductive PEDOT:PSS hydrogel through subsequent swelling in a wet environment (Figure 3c). To showcase the achievement of high-resolution microscale printing, they used a 7wt% PEDOT:PSS nanofibril-based conducting polymer ink to create intricate mesh patterns through nozzles of various diameters: 200 µm, 100 µm, 50 µm, and 30 µm (referred to as Figure 3d–g). The conducting polymer ink’s favorable rheological properties also facilitate the construction of multi-layered microstructures with high aspect ratios, using a 100 µm nozzle and incorporating up to 20 layers. 

Figure 3h illustrates the step-by-step progression of creating a 20-layered meshed structure using the conducting polymer ink. Following the printing, Figure 3i displays the 3D-printed conducting polymer mesh after undergoing a dry-annealing process, which is crucial for enhancing its conductivity. In contrast, Figure 3j showcases the same 3D-printed conducting polymer mesh but in its hydrogel state, highlighting the adaptability of this material to different environmental conditions. Figure 3k provides a visual representation of the sequential snapshots for 3D-printing overhanging features across high aspect ratio structures using the conducting polymer ink, demonstrating the ink’s ability to maintain structural integrity during complex printing. Finally, Figure 3l reveals the 3D-printed conducting polymer structure with overhanging features, which has been transformed into a hydrogel state. Collectively, these figures demonstrate the versatility, resilience, and transformative capabilities of conducting polymer inks in advanced 3D printing applications.

#### 2.1.3. Hybrid Hydrogels

Apart from natural and synthetic hydrogels, hybrid hydrogels can also be prepared by combining the components from both natural and synthetic sources, leveraging the advantages of each material type. Hybrid hydrogels often strike a balance between natural and synthetic hydrogels, aiming to improve printability by combining the advantages of both material types. They can also harness the biocompatibility of natural components while introducing controlled synthetic elements to enhance mechanical properties and stability. Also, these hydrogels aim to combine the mechanical advantages of synthetic materials with the biocompatibility of natural materials, offering a balanced approach for applications requiring both properties. Tang et al. [29] delved into the domain of hybrid hydrogels for three-dimensional cell culture and employed stereolithography to craft nanocellulose/PEGDA aerogel scaffolds, thereby introducing tunability to the modulus of these structures. This investigation is of notable significance within the realms of tissue engineering and regenerative medicine as it provides a platform for 3D cell cultivation, effectively emulating natural tissue environments. Cao and their collaborators [30] have also explored the potential of antibacterial hybrid hydrogels as a robust tool for combatting microbial infections. This research represents a vital area of study with broad-reaching implications in the field of medicine, where the development of infection-resistant materials can significantly elevate patient care and mitigate the transmission of infectious diseases.

Palmese et al. [31] provided a comprehensive investigation and multifaceted applications of hybrid hydrogels within the biomedical field, which underscores their versatility and potential, showcasing the diverse ways they can be applied to address various medical challenges, from developing drug delivery systems to advancing tissue engineering. Vasile and his research team also examined the incorporation of natural polymers into hybrid hydrogels for medical applications [32]. Their study sheds light on the diverse utility of these materials in healthcare, indicating their roles in drug delivery, wound healing, and various other medical interventions. This research emphasizes the increasing interest in harnessing the properties of natural polymers within medical contexts. Liao et al. [33] introduced a unique hybrid hydrogel with the specific aim of preventing bone tumor recurrence and promoting bone regeneration. This specialized hydrogel incorporates gold nanorods and nanohydroxyapatite and utilizes photothermal therapy. This innovative approach holds great promise for addressing bone-related health concerns and managing bone tumors, offering a novel and effective treatment strategy. Anisotropic hybrid hydrogels with mechanical properties reminiscent of tendons or ligaments werealso explored [34] for tissue engineering, providing the potential to mimic the mechanical characteristics of specific tissues. This development offers a valuable resource for creating artificial ligaments and tendons. Yang et al. [35] introduced 3D macroporous oxidation-resistant Ti_3_C_2_Tx MXene hybrid hydrogels, which demonstrate remarkable supercapacitive performance with an extraordinarily long cycle life. This research holds significant implications for energy storage applications, particularly in supercapacitors, where long-term stability and performance are imperative. Collectively, these studies contribute to the expanding field of hybrid hydrogels, underscoring their potential and versatility in a wide range of biomedical and materials science applications. These applications span from healthcare to advanced materials used in sustainable technologies. Table 1 comparatively summarizes the various properties of natural, synthetic, and hybrid hydrogels in terms of preparation, printability, biocompatibility, and mechanical properties.

#### 2.1.4. Crosslinked Hydrogels

Depending upon crosslinking, hydrogels can be classified as physically crosslinked hydrogels and chemically crosslinked hydrogels. Physically crosslinked hydrogels are formed through non-covalent interactions, such as hydrogen bonding or physical entanglements, and are reversible and responsive to environmental changes, while chemically crosslinked hydrogels are formed by covalent bonds between polymer chains, resulting in higher stability and mechanical strength. Examples include photopolymerization and chemical crosslinking agents. Iwanaga et al. [36] focused their efforts on designing and producing fully developed engineered pre-cardiac tissue using 3D bioprinting techniques coupled with enzymatic crosslinking hydrogels. This pioneering investigation shows great potential in the fields of regenerative medicine and cardiac tissue engineering, addressing the significant hurdle of generating functional heart tissues. Recently, Ianchis et al. [37] also delved into creating innovative green crosslinked hydrogels derived from salecan and initiated their preliminary exploration of these hydrogels in the context of 3D printing. The prospects of utilizing sustainable and environmentally friendly hydrogel materials in additive manufacturing processes havebeen highlighted in this research, with implications spanning drug delivery and tissue engineering. Another groundbreaking work by Farsheed and his collaborators has been witnessed in the realm of 3D printing of self-assembling nanofibrous multidomain peptide hydrogels [38]. This research has the potential to revolutionize the fabrication of intricate and adaptable nanofibrous structures through 3D printing, introducing novel applications in regenerative medicine, drug delivery, and tissue engineering.

### 2.2. Properties of Hydrogels

Hydrogels showcase a myriad of captivating features, notably their substantial water content, customizable mechanical attributes, swelling dynamics, biocompatibility, permeability, and more. With water constituting up to 90% of their composition, hydrogels offer a conducive environment for interactions with biological systems. Their mechanical properties, encompassing factors like stiffness, elasticity, and resilience, are finely adjustable through parameters such as polymer type, crosslinking density, and composition. The responsive nature of hydrogels, exhibiting swelling and deswelling reactions to environmental shifts like pH and temperature, positions them as ideal candidates for controlled drug delivery systems. Many hydrogels exhibit biologically inert and non-toxic traits, ensuring their safe utilization in medical applications without adverse effects on cells and tissues. Moreover, their ability to selectively allow the diffusion of small molecules while impeding larger ones holds promise for applications in separation and filtration processes. 

The unique characteristics, diverse compositions, and tunable crosslinking methods of hydrogels make them invaluable materials with wide-ranging applications across biomedical, pharmaceutical, and various other industries, fostering innovation and influencing multiple aspects of contemporary life. Table 2 provides a summary of the various properties of hydrogels along with their corresponding applications. While biomedical and pharmaceutical applications of hydrogel materials are notably prominent, their extensive promise in tissue engineering stands out. Hydrogels, owing to their biocompatibility and resemblance to natural tissues, serve as scaffolds for tissue regeneration, facilitating the repair of damaged tissues and organs. In drug delivery systems, hydrogels play a pivotal role by enabling the controlled release of therapeutic agents over time. Their capacity to encapsulate drugs and proteins ensures targeted delivery, minimizing side effects and enhancing treatment outcomes. 

### 2.3. Designing Hydrogel Formulations for 3D Printing

Designing hydrogel formulations for 3D printing involves optimizing various parameters to ensure proper printability, structural integrity, and compatibility with the chosen printing method. A delicate balance between rheological properties, gelation kinetics, and print fidelity is highly imperative for designing hydrogel formulations for 3D printing [39]. Rheological characteristics determine the hydrogel’s flow behavior during printing, while gelation kinetics influence the temporal aspects of the printing process. Achieving optimal print fidelity requires a careful balance of these properties to ensure accurate layer deposition and the overall success of 3D-printed hydrogel constructs, particularly in biomedical applications. Additives also play a critical role in tailoring hydrogel properties to meet specific application requirements, enabling the creation of structures with optimized mechanical, biological, and functional characteristics. 

It should be noted here that rheology is the study of the flow and deformation of materials. In the context of hydrogels for 3D printing, rheological properties such as viscosity and shear-thinning behavior are crucial. A suitable viscosity ensures proper extrusion through the printer nozzle, while shear-thinning behavior allows the hydrogel to reduce viscosity under shear stress during printing, facilitating smooth flow and layer deposition. Several researchers have shed light on the correlation between rheological properties and printing behavior. Bom et al. [39] explored this correlation, emphasizing the importance of understanding rheological properties in achieving successful 3D printing outcomes. Amorim et al. [40] provided insights into the shear rheology of inks for extrusion-based 3D bioprinting, contributing to the fundamental understanding of the behavior of printing materials under shear stress. In a related vein, Kokol et al. [41] investigated how flow- and horizontally-induced cooling rates during 3D cryo-printing affect the rheological properties of gelatine hydrogels. Kim et al. [42] focused on enhancing the rheological behaviors of alginate hydrogels with carrageenan, specifically for extrusion-based bioprinting applications. Townsend et al. [43] emphasized the significance of precursor rheology for hydrogel placement in medical applications and 3D bioprinting, underscoring the importance of flow behavior before crosslinking. Zhou et al. [44] introduced microbial transglutaminase-induced controlled crosslinking of gelatin methacryloyl, showcasing a method to tailor rheological properties for optimized 3D printing.

On the other hand, gelation refers to the process by which a liquid hydrogel transforms into a gel, solidifying the material. Gelation kinetics in 3D printing involves the study of the time taken for the hydrogel to transition from a liquid to a solid state. The speed of gelation is a critical factor influencing the printing speed and the overall printing time. Ideally, the gelation kinetics should match the printing speed to ensure proper layer-by-layer deposition. Apart from these, print fidelity encompasses the accuracy and precision of the printed structure compared to the digital design. Several factors contribute to print fidelity, including the rheological properties of the hydrogel, the gelation kinetics, and the overall printing parameters. Achieving high print fidelity is crucial for accurately replicating complex structures, especially in applications like tissue engineering, where the precise arrangement of cells and biomaterials is essential for functionality. Schwab and colleagues [45] delved into the printability and shape fidelity of bioinks, offering comprehensive insights crucial for developing precise 3D bioprinted structures. On the other hand, Mora-Boza et al. [46] explored the potential of glycerylphytate as an ionic crosslinker for 3D printing, demonstrating its effectiveness in producing multi-layered scaffolds with enhanced shape fidelity and desirable biological features. In a different avenue, Huang et al. [47] investigated the incorporation of bacterial cellulose nanofibers to improve the stress and fidelity of 3D-printed silk-based hydrogel scaffolds, introducing a novel approach to scaffold enhancement. Sheikhi et al. [48] focused on the 3D printing of jammed self-supporting microgels, presenting an alternative mechanism that addresses shape fidelity, crosslinking, and conductivity. Table 3 summarizes various key parameters and properties under rheology, gelation kinetics, print fidelity, and their effects on 3D printing processes.

## 3. 3D Printing Methods for Hydrogels

Several 3D printing methods are employed to create hydrogel-based structures, each with its own set of advantages and limitations. In selecting a 3D printing method for hydrogel-based projects, the trade-offs between resolution, speed, and material compatibility should be carefully considered to align with the desired application and project requirements. The various 3D printing technologies include extrusion-based 3D printing, inkjet-based printing, stereolithography (SLA), laser-assisted bioprinting, digital light processing (DLP), etc. [6,7,8,9,10]. Figure 4 schematically illustrates the various 3D printing technologies that can be successfully employed for hydrogel printing. Kantaros et al. [49] explored the technologies and resources employed in 3D printing within the realm of regenerative medicine and shed light on the instrumental tools and methodologies used to propel the field forward, facilitating the generation of intricate tissue constructs for transplantation and mending. Bedell and colleagues [50] investigated human gelatin-based composite hydrogels tailored for osteochondral tissue engineering, adapting them into bioinks suitable for a variety of 3D printing techniques. Their investigation contributes to the advancement of novel materials and printing methodologies aimed at restoring damaged joints and cartilage. The emerging domain of 3D printing for immobilizing biocatalysts has also been reported [51,52]. This study underscores the immense potential of 3D printing in biotechnology, enabling the efficient immobilization of enzymes for diverse applications, spanning bioprocessing and biofuel production.

### 3.1. Extrusion-Based Printing

Extrusion-based 3D printing for hydrogels is a sophisticated additive manufacturing process with immense potential in various fields, particularly biomedicine. This method harnesses the unique properties of hydrogel materials, which consist of a water-based gel with a polymer network, making them suitable for extrusion. A hydrogel material is meticulously prepared in this process, often as a viscous bioink, customized with specific additives like cells or therapeutic agents. A specialized extruder nozzle is employed in a 3D printer, precisely depositing the hydrogel layer by layer to construct intricate three-dimensional structures. Each layer is carefully deposited and, if needed, undergoes a crosslinking process to solidify and stabilize the structure. This technique is widely used in biomedicine to fabricate tissue scaffolds, organ models, and drug delivery systems and in other industries, such as food and cosmetics, for creating customized products. In recent years, a series of notable studies have significantly advanced the field of extrusion-based 3D printing, introducing innovative materials and technologies that broaden its application horizon. Cheng et al. [53] delved into the printability of a cellulose derivative for 3D printing, specifically focusing on its use as a biodegradable support material. Itmarkeda vital step toward more sustainable and versatile 3D printing processes, which can contribute to environmentally friendly manufacturing. Zhou et al. [54] explored the potential of gelatin-oxidized nanocellulose hydrogels in extrusion-based 3D bioprinting for tissue engineering and regenerative medicine, introducing a novel avenue in the quest for innovative and regenerative biomaterials.

Dong et al. [55] have demonstrated the extrusion 3D printing of a gelatine methacrylate/Laponite nanocomposite hydrogel, emphasizing its promise for applications in the realm of bone tissue regeneration. A unique approach by exploring advanced printable hydrogels derived from pre-crosslinked alginate was demonstrated by Falcone et al. [56]. This study highlights the suitability of these materials for semi-solid extrusion 3D printing, effectively extending the capabilities of this technology in creating intricate and complex structures for a wide array of applications, from tissue engineering to drug delivery systems. The work of Murphy et al. [57] introduced a fascinating twist by focusing on 3D extrusion printing of stable constructs composed of photoresponsive polypeptide hydrogels. This innovative approach allows for the creation of responsive and tunable 3D-printed materials, paving the way for a variety of applications where dynamic, light-responsive materials are required. Hu et al. [58] explored the extrusion 3D printing of a cellulose hydrogel skeleton, which opens new possibilities for creating eco-friendly and sustainable materials through 3D printing. Their work aligns with the growing demand for more environmentally responsible manufacturing practices. Substantial progress in advancing extrusion 3D bioprinting, with a specific focus on multicomponent hydrogel-based bioinks, was also demonstrated [59]. By pushing the boundaries of complexity and functionality in 3D-printed constructs, their research directly contributes to developing cutting-edge biomedical applications, from tissue engineering to personalized medicine.

Apart from experimental works, simulations for the extrusion 3D printing of chitosan hydrogels have also been reported [60]. Their research provides valuable insights into the virtual design and optimization of bioprinted structures, streamlining the development of precise and complex geometries for tissue engineering and other biomedical applications. Table 4 summarizes various extrusion-based 3D-printed hydrogels for diverse applications. Extrusion-based 3D printing for hydrogels offers versatility, cost-effectiveness, and the ability to work with various hydrogel formulations, making it a valuable tool for research, development, and manufacturing applications. However, moderate to lower resolution compared to other methods, potential nozzle clogging, and slower printing speeds due to layer-by-layer deposition limit its utility. 

### 3.2. Inkjet-Based Printing

Inkjet-based 3D printing for hydrogels is a cutting-edge additive manufacturing technique that harnesses the precision of inkjet printing to construct three-dimensional structures using hydrogel materials. This innovative method is particularly well-suited for hydrogels, which often possess high water content and distinct rheological properties. The process begins with preparing a hydrogel ink tailored to the specific application, often containing crosslinking agents, water, and additives like cells or therapeutic compounds. Specialized inkjet 3D printers are equipped with printheads designed to accommodate these hydrogel inks. These printheads feature micro-sized nozzles that dispense minute droplets of the hydrogel ink onto a build platform. Each droplet represents a pixel; layer by layer, these droplets create the desired three-dimensional object. Following deposition, the hydrogel droplets are solidified, typically through methods such as UV light exposure or temperature adjustment. The technology is versatile, finding applications in biomedicine for creating tissue constructs, implants, and precise drug delivery systems. Additionally, it extends to various industries, enabling the customization of products with hydrogel components. In the field of 3D bioprinting, Suntornnond and colleagues [61] focused on improving the printability of hydrogel-based bioinks for thermal inkjet bioprinting applications and employed saponification and heat treatment processes to enhance the performance of these bioinks, aiming to optimize their application in creating complex biological structures.

Inkjet-based 3D printing technology also plays a vital role in fabricating microfluidic devices [62,63]. In the realm of advanced manufacturing and micro-optofluidic applications, Saitta et al. [64] employed a regression approach to model the refractive index measurements of innovative 3D-printable photocurable resins, a crucial step in micro-optofluidic applications. Marzano et al. [65] explored the potential of 3D printers and UV-cured optical adhesives in creating V-shaped plasmonic probes tailored for medical applications. Adamski et al. [66] have introduced an innovative technique for DNA analysis employing a lab-on-a-chip (LOC) system created through inkjet printing and on-chip gel electrophoresis. This novel capillary gel electrophoresis method is designed to separate genetic materials, primarily DNA and RNA, by examining the migration velocities of their fractions. The electrophoretic chip itself was produced using inkjet 3D printing. Two methods for fabricating the microfluidic electrophoretic chip are schematically depicted in Figure 5a,b. In the glass microengineering approach (Figure 5a), two glass substrates were bonded together to create the structure, including the injector and separation microchannels. Conversely, in the inkjet 3D printing method (Figure 5b), the entire structure was created in a single printing process on a chip. The LOC structure, as fabricated, encompasses an injection microchannel, two separation microchannels with lengths of 20 mm and 50 mm, and an integrated buffer and sample reservoir components. The chip’s CAD design is presented in Figure 5c, which was then exported to a geometric file (Figure 5d), leading to the final structure illustrated in Figure 5e. To analyze DNA, the electrophoretic gel was injected into the chip’s injection microchannel, subsequently entering the separation microchannel (Figure 5f–h). Finally, it underwent optical detection employing fluorescence modulation, accomplished with laser light and a CCD mini-camera, as shown in Figure 5i.

Recent advancements in 3D inkjet printing technology have paved the way for fabricating intricate, cell-laden hydrogel structures, opening up new possibilities in tissue engineering, regenerative medicine, and drug delivery [67,68]. Teo et al. [69] contributed to this field by enabling the creation of free-standing 3D hydrogel microstructures through microreactive inkjet printing. Meanwhile, Nakagawa et al. [70] explored the use of star block copolymer hydrogels cross-linked with various metallic ions, demonstrating the versatility of inkjet printing in creating complex structures. In a study by Jiao et al. [71], inkjet printing was employed to fabricate alginate/gelatin hydrogels with tunable mechanical and biological properties, expanding the potential applications of these materials.

Yoon et al. [72] introduced an inkjet–spray hybrid printing approach for the 3D freeform fabrication of multilayered hydrogel structures, showcasing a versatile technique for complex hydrogel architectures. Peng et al. [73] demonstrated the surface patterning of hydrogels using ion inkjet printing, enabling programmable and complex shape deformations. Additionally, Duffy et al. [74] presented a 3D reactive inkjet printing method for poly-ɛ-lysine/gellan gum hydrogels, showing promise for potential corneal constructs. Table 5 summarizes various inkjet-based 3D-printed hydrogels for diverse applications.

A comparative study involving hyaluronic acid hydrogel exquisite micropatterns using two distinct methods, photolithography and light-cured inkjet printing, to create intricate and precise microstructures has been reported by Chen et al. [75]. This research contributes to developing advanced techniques for fabricating microscale patterns with hyaluronic acid hydrogels, which are relevant in various biomedical and tissue-engineering applications. While inkjet-based 3D printing for hydrogels offers high resolution and intricate control, it also presents challenges related to material viscosity and nozzle maintenance. Nonetheless, it holds great promise for advancing research and development in fields demanding precise, biocompatible structures.

### 3.3. Stereolithography (SLA)

Stereolithography-based 3D printing for hydrogels represents a cutting-edge additive manufacturing technique that harnesses the power of light to create intricate three-dimensional structures using hydrogel materials. This technology builds upon the principles of traditional stereolithography but adapts them to accommodate the unique properties of hydrogels, which typically consist of water-based polymer networks. The process begins with a hydrogel resin that may contain photo-initiators, crosslinkers, and other additives. In this method, a precisely controlled light source, often in the form of a laser or projector, is used to selectively solidify specific regions of the hydrogel resin. As each layer is solidified, the build platform descends incrementally, allowing for the gradual construction of the 3D object. Stereolithography offers exceptional precision and fine detail, making it wellsuited for applications in tissue engineering, biomedical devices, and microfluidic systems. Post-printing, additional steps such as rinsing and curing may be required to enhance structural stability. In the realm of 3D printing, various studies have explored the application of stereolithography to create hydrogels with unique properties and capabilities. Kalossaka and colleagues [76] delved into the creation of 3D nanocomposite hydrogels featuring lattice vascular networks, demonstrating the potential for engineering intricate structures using this technique. Karakurt et al. [77] focused on SLA 3D printing of hydrogels loaded with ascorbic acid, investigating their controlled release properties. The study sheds light on the use of SLA in drug delivery systems.

The impact of SLA 3D printing on the properties of PEGDMA hydrogels, contributing insights into the interactions between the printing process and hydrogel characteristics, has also been discussed [78]. Magalhães et al. [79] explored the use of low-cost stereolithography technology to create 3D hydrogel structures, offering potential applications in biomedicine and biomaterials. Effects of PEGDA photopolymerization in micro-stereolithography on 3D-printed hydrogelstructures, addressing aspects of structural integrity and swelling behavior, havebeen reported by Alketbi et al. [80]. Sun et al. [81] introduced a new stereolithographic 3D printing strategy for hydrogels, focusing on achieving a large mechanical tunability and self-weldability, which opens doors to more versatile and functional hydrogel applications. Table 6 summarizes various stereolithography-based 3D-printed hydrogels for diverse applications. 

Stereolithography-based 3D printing for hydrogels is celebrated for its ability to create complex, high-resolution structures, but challenges such as limited material options and the need for specialized equipment are considerations in its implementation. Nevertheless, it remains a powerful tool in the field of hydrogel-based additive manufacturing, enabling the fabrication of advanced biomimetic structures for a wide range of applications. 

### 3.4. Digital Light Processing (DLP)

DLP-based 3D printing for hydrogels is an innovative and precise additive manufacturing technique that harnesses digital light projection to construct intricate 3D structures using hydrogel materials. This method builds upon the principles of photopolymerization, where a liquid resin containing hydrogel components is selectively cured by a digital light source, typically a projector or UV LED. This process proceeds layer by layer, with each layer being solidified by the precise projection of light patterns, forming the desired object. DLP 3D printing for hydrogels is celebrated for its remarkable speed, high resolution, and ability to create complex geometries with fine detail. It finds particular utility in biomedical applications, including tissue engineering and the fabrication of customized implants and drug delivery systems. 

Recent advancements in the DLP 3D printing of hydrogels have ushered in a new era of possibilities and applications. Several research groups have made significant contributions to this evolving field. Hosseinabadi and collaborators [82] delved into the critical aspects of ink material selection and optical design in DLP 3D printing, underlining the importance of precision and performance in hydrogel printing. A concise review summarizing the key developments and prospects of this technology, providing an overview of the current state of DLP 3D printing for hydrogels, has been presented by Ding et al. [83]. Sun et al. [84] introduced a novel approach to DLP 3D printing by presenting hydrogels with hierarchical structures enhanced through lyophilization and ionic locking, offering new possibilities for tailored hydrogel properties. In a different direction, Dong and colleagues [85] focused on creating tough supramolecular hydrogels using DLP 3D printing, emphasizing their potential application as impact-absorption elements. An effective DLP 3D printing strategy was also introduced for cellulose hydrogels with high strength and toughness, wellsuited for strain sensing applications [86]. Cafiso et al. [87] explored 3D printing of fully cellulose-based hydrogels through DLP, highlighting the sustainability and versatility of this approach. On the biomedical front, Wang et al. [88] successfully fabricated antimicrobial hydrogels using DLP 3D printing, leveraging sustainable resin and hybrid nanospheres for potential applications in healthcare.

Caproili and colleagues [89] introduced a novel 3D-printed hydrogel imbued with self-healing properties, employing readily available materials and a DLP printer commonly found in the market. The ingenious design of this system involved creating a sequential semi-interpenetrated network by introducing chemical covalent network precursors into a solution containing a linear polymer. Following polymerization, this linear polymer is effectively encapsulated within the cross-linked matrix. The photocurable ink formulation involved the blending of an aqueous solution of unmodified, non-crosslinked Poly (vinyl alcohol) (PVA) with acrylic acid (AAc), the cross-linking agent Poly (ethylene glycol) diacrylate (PEGDA), and a water-compatible photoinitiator based on diphenyl (2,4,6-trimethylbenzoyl)phosphine oxide (TPO). Figure 6a provides a visual representation of the chemical structures of the initiator, monomer, cross-linker, and mending agent within the photocurable resin. Figure 6b illustrates the creation of the semi-interpenetrated network: a physical network is blended with the precursors of the chemical network, which materializes during exposure to light. The outcome is a hydrogel composed of PVA chains that are uniformly distributed and incorporated into a cross-linked acrylic matrix. For tensile testing, two distinct specimens were produced, each featuring a different color, one printed with methyl red sodium salt dye and the other with brilliant green dye. These samples demonstrated their capacity to endure bending deformation immediately upon rejoining, as depicted in Figure 6c. Figure 6d showcases a perforated cylindrical structure printed with methyl red sodium salt dye, clearly demonstrating that the reconnected sample could withstand stretching deformation after a 2h healing process. Figure 6e reveals body-centered cubic lattice-like structures printed with both methyl red sodium salt dye and brilliant green dye, enhancing visual comprehension. The inset in this figure vividly portrays diffusion at the interface after 12 h of contact, owing to the gradient of the dye. Figure 6f,g display 3D fabricated samples with a PVA 0.8 formulation, featuring a body-centered cubic lattice-like structure printed with methyl red sodium salt dye and an axis-symmetric structure with a central pillar printed with brilliant green dye, respectively. Finally, Figure 6h offers insight into the stress–strain curves of the self-healed hydrogels, showcasing their performance over increasing healing durations.

Zhang et al. [90] experimented with Antheraea pernyi silk fibroin bioinks for DLP 3D printing, offering new opportunities in bioprinting and regenerative medicine. In another innovative stride, Xiang et al. [91] focused on creating 3D-printed high-toughness double network hydrogels via DLP, contributing to the development of mechanically robust hydrogel materials. Lopez-Larrea et al. [92] explored the application of PEDOT-based photopolymerizable inks for biosensing through DLP 3D printing, marking the potential for enhanced biosensor development. Table 7 summarizes various DLP-based 3D-printed hydrogels for diverse applications. 

It is noteworthy to mention here that in the case of DLP-based 3D printing technology, post-processing steps, including rinsing and additional curing, may be required after printing to enhance structural stability. However, it is important to note that material selection and compatibility with the photopolymerization process are key considerations. Despite these challenges, DLP-based 3D printing stands as a powerful tool for creating intricate hydrogel structures, pushing the boundaries of precision and complexity in various scientific and medical fields. Table 8 summarizes the principle and various properties of 3D printing technologies, including extrusion-based, inkjet-based, SLA, and DLP.

## 4. Applications of 3D-Printed Hydrogels

### 4.1. Role of a 3D-Printed Hydrogel in Biomedical Applications

Hydrogel 3D printing has emerged as a groundbreaking technology in the field of biomedical engineering, offering innovative solutions in tissue engineering, regenerative medicine, and drug delivery [93,94,95,96,97]. By harnessing the unique properties of hydrogels, this approach enables the creation of precise and complex structures that closely mimic the natural extracellular matrix. By creating intricate structures with tailored properties, this technology opens avenues for personalized medicine, faster wound healing, and the development of advanced organ-on-a-chip systems, ultimately reshaping the landscape of biomedical research and healthcare. 

#### 4.1.1. Tissue Engineering

Hydrogel 3D printing plays a pivotal role in fabricating scaffolds and constructs for tissue engineering, providing a supportive framework for cells to adhere, proliferate, and differentiate, ultimately aiding in regenerating damaged tissues or organs. Lan et al. [98] provided a comprehensive review of progress in 3D printing for bone tissue engineering, highlighting the potential for personalized and effective solutions in orthopedics. Varaprasad et al. [99] underscored the importance of alginate as a biomacromolecule for 3D-printing hydrogels in biomedical applications, focusing on its potential for creating bioactive scaffolds. 

Xu et al. [100] presented a novel glucose-responsive antioxidant hybrid hydrogel designed to enhance diabetic wound repair, highlighting its potential in addressing critical healthcare challenges. Tsegay and colleagues [101] provided a review on smart 3D-printed hydrogelskin wound bandages, emphasizing the potential of these advanced wound dressings for improved patient care and recovery. Hydrogel 3D printing is also used to create microfluidic devices that mimic the structure and function of human organs, allowing researchers to study diseases and drug responses in a controlled environment. These systems provide insights into organ-level behavior and drug efficacy without the need for animal testing. Hydrogel-based constructs can be engineered with vascular networks, enabling the perfusion of nutrients and oxygen to cells deep within the tissue. This advancement is crucial for creating thicker tissues and organs, addressing one of the challenges in tissue engineering.

Guo et al. [102] introduced a groundbreaking 3D liver-inspired detox device. It is madeusing3D-printing hydrogels with special nanoparticles that attract, capture, and sense toxins. This innovative system efficiently traps toxins using a liver-like microstructure. Their work shows that this device effectively neutralizes toxins, offering a promising path for advanced detoxification platforms. Leveraging the benefits of a 3D biomimetic structure for enrichment, separation, and detection, a bioinspired 3D detoxification device has been devised. This innovative device incorporates polydiacetylene (PDA) nanoparticles into a precisely designed 3D matrix resembling a modified liver lobule configuration, achieved through advanced 3D printing technology known as dynamic optical projection stereolithography (DOPsL). DOPsL employs a digital mirror array device (DMD) to create dynamic photomasks, allowing for the layer-by-layer photopolymerization of biomaterials into intricate 3D structures.

Figure 7a visually illustrates the integration of PDA nanoparticles (green) within a PEGDA hydrogel matrix (grey) featuring a liver-mimicking 3D structure produced through 3D printing. This structure can attract, capture, and detect toxins (red), while the modified liver lobule-like matrix efficiently traps these harmful substances. This biomimetic 3D detoxifier holds promise for clinical applications, serving as an effective means of collecting and removing toxins. Furthermore, they have successfully immobilized toxins within 3D-printed hydrogel nanocomposites, each sporting distinct surface patterns in the form of flower-like shapes with varying diameters (large, medium, and small). When exposed to a melittin solution, as depicted in Figure 7b–d, the red fluorescence indicates interactions between PDA and melittin, offering insights into toxin localization.

Hydrogel-based dressings with 3D-printed structures provide an ideal environment for wound healing. These dressings can maintain a moist and protective environment, promote cell migration, and facilitate tissue regeneration. For example, 3D-printed hydrogel dressings can cover burn wounds, providing a barrier against infection, enhancing wound healing, and reducing pain and scarring. Hydrogel-based wound dressings can be infused with growth factors and antimicrobial agents, accelerating healing in chronic wounds like diabetic ulcers. Xiong et al. [103] fabricated DLP-based 3D-printed wearable sensors possessing self-adhesion and self-healing properties using polymerizable rotaxane hydrogels. In their study, acrylated β-cyclodextrin combined with bile acid is self-organized into a polymerizable pseudorotaxane through precise host–guest recognition. This pseudorotaxane is subsequently photopolymerized alongside acrylamide to create conductive polymerizable rotaxane hydrogels (PR-Gel). PR-Gel exhibited robust self-adhesion to the skin, as depicted in Figure 8a, and notably, theyleft no residue or caused any allergic reactions when adhered to the wrist. Lap-shear tests were conducted to assess their adhesion to various surfaces, including nitrile gloves, glass plates, silicone rubber, aluminum, and pigskin (Figure 8b). Although the adhesion strength between PR-Gel and these substrates fell within the range of 4.5 to 6.0 kPa, which is lower than that of some highly adhesive hydrogels, this moderate self-adhesion ensured a secure bond between PR-Gel and human skin, as demonstrated in Figure 8a. This feature guarantees stable signal transmission and consistent adhesion for wearable sensor applications. To showcase the self-healing capabilities, two cylindrical PR-Gel samples were cut and then reassembled at the incision point, serving as a qualitative demonstration. For better visual distinction, one of the samples was stained in a red-orange hue, as depicted in Figure 8c. These mended gels exhibited resilience to tensile deformation without fracturing at room temperature. The self-healing prowess of PR-Gel underwent further scrutiny through structural damage and recovery assessments involving continuous strain sweeps with alternating low (1%) and high (500%) oscillatory excitations. In Figure 8d, the storage modulus (G′) and loss modulus (G″) of PR-Gel during continuous strain sweeps illustrate theirrapid self-healing properties post-damage. This process could be repeated multiple times, underscoring the reproducible restoration of the PR-Gel network.

#### 4.1.2. Regenerative Medicine

The 3D-printed hydrogels have emerged as a transformative technology with significant implications for regenerative medicine. By combining the precision of 3D printing with the versatility of hydrogels, researchers and clinicians can fabricate intricate structures that mimic the complexity of biological tissues. This innovative approach enables the creation of customized scaffolds tailored to match the specific geometry and mechanical properties of target tissues. The hydrogel’s biocompatibility and ability to encapsulate cells make it an ideal candidate for supporting cellular growth and tissue regeneration. Moreover, the controlled release of bioactive molecules from these 3D-printed hydrogels further enhances their regenerative potential. Applications span a wide range, from engineering bone and cartilage to creating vascularized tissues. As the field advances, 3D-printed hydrogels hold promise for revolutionizing regenerative medicine, offering new avenues for personalized and effective tissue repair and regeneration. Tajik et al. [97] critically reviewed the 3D printing of hybridhydrogel materials for tissue engineering, underlining the significance of this approach in advancing regenerative medicine.

Deptula et al. [104] reported various diverse applications of 3D-printed hydrogels in the domains of wound healing and regenerative medicine. Notably, their research highlights the creation of skin wound bandages through 3D-printed hydrogels and the integration of these hydrogels into tissue-engineering strategies. The work of Bhatnagar et al. [105] emphasized the significance of hydrogels in regenerative medicine, showcasing their potential for tissue repair and regeneration. Nanoengineered biomimetic hydrogels, a major advancement detailed by Cernencu et al. [106], have enabled the fabrication of 3D-printed constructs, offering a promising avenue for regenerative medicine applications. Natural hydrogels, as reviewed by Catoira et al. [107], contribute to the overview of materials used in regenerative medicine, emphasizing their relevance in the field. The comprehensive exploration of hydrogels in regenerative medicine, including their applications and challenges, is presented in the work by Hasirci et al. [108]. Aghamirsalim et al. [109] delved into the specific use of 3D-printed hydrogels for ocular wound healing, showcasing the versatility of this technology. Shamma et al. [110] focused on the use of triblock copolymer bioinks, specifically pluronic F127, in hydrogel 3D printing for regenerative medicine. Diverse applications of hydrogels in regenerative medicine and other biomedical fields have also been reviewed [111]. The engineering of hydrogels for personalized disease modeling and regenerative medicine wasdiscussed by Tayler and Stowers [112], highlighting the potential for tailored therapeutic approaches. Heo et al. [113] presenteda remarkable application, where 3D-printed microstructures incorporated with hybrid nano hydrogels enhance bone tissue regeneration. Foyt et al. [114] focused on exploiting advanced hydrogel technologies to address key challenges in regenerative medicine, showcasing the continuous evolution of this field.

#### 4.1.3. Drug Delivery

The 3D-printed hydrogelalso offers precise control over the spatial distribution of drug-loaded hydrogel matrices. This technology enables the design of drug delivery systems that release therapeutic agents in a controlled and sustained manner, improving treatment efficacy and reducing side effects. Hydrogel-based implants can be 3D printed with encapsulated drugs or growth factors, allowing for site-specific and controlled release. This approach is used in cancer therapy, wound healing, and bone regeneration. Oral drug delivery systems can also be created using a 3D-printed hydrogelthat releases drugs gradually, enhancing bioavailability and reducing the frequency of dosing. Patient-specific drug formulations can also be achieved through 3D printing, tailoring drug release profiles to individual patient needs, which is particularly relevant in cases of chronic diseases.

Tetyczka et al. [100] examined the integration of itraconazole nanocrystals on hydrogel contact lenses via inkjet printing, demonstrating the implications of this technique for ophthalmic drug delivery. Maiz-Fernández et al. [115] introduced self-healing hyaluronic acid/chitosan polycomplex hydrogels with drug release capabilities, opening avenues for customized and responsive medical interventions. Larush et al. [116] explored the potential of 3D printing responsive hydrogels for drug delivery, emphasizing the versatility of this approach. Aguilar-de-Leyva et al. [117] delved into the development of 3D-printed drug delivery systems using natural products, aligning with the growing trend toward sustainable and bio-inspired pharmaceutical solutions. Martinez et al. [118] focused on stereolithographic 3D printing to fabricate drug-loaded hydrogels, presenting a precise and controlled drug-delivery platform. NIR-triggered drug release from 3D-printed hydrogel/PCL core/shell fiber scaffolds, showcasing the potential for localized cancer therapy and wound healing, has also been reported [119,120]. Dreiss [121] provided insights into hydrogel design strategies for drug delivery, offering a comprehensive overview of the evolving landscape in this domain. Wang et al. [122] contributed to the field by 3D printing shape memory hydrogels with internal structures for drug delivery, adding a layer of sophistication to the design of drug-release systems.

### 4.2. Hydrogel 3D Printing across Industries

While hydrogel 3D printing has gained significant traction in biomedical applications, its versatility extends to diverse industries beyond healthcare. The unique properties of hydrogels, such as high water content and tunable mechanical properties, make them intriguing candidates for various applications in industries like food, cosmetics, and electronics. 

#### 4.2.1. Food Industry

In the food industry, 3D-printed hydrogels offer a range of applications, revolutionizing the way we produce and consume food. This technology enables the creation of customized food structures with unique textures and shapes while also allowing for precise control over the release of nutrients and flavors. The ability to tailor mechanical properties enhances the overall eating experience, mimicking desired textures in food products. Moreover, 3D printing facilitates the integration of functional ingredients, such as probiotics and antioxidants, in a spatially controlled manner. The technology’s potential to reduce food waste, streamline processing steps, and contribute to personalized nutrition makes it a promising innovation in the quest for sustainable and innovative food solutions. Chefs can leverage 3D printing for artistic culinary designs, while companies benefit from rapid prototyping for quicker product development. 

In the rapidly evolving landscape of food technology, hydrogels derived from natural polymers have emerged as versatile materials with transformative applications. Zhang et al. [123] extensively explored the diverse applications of natural polymer-based hydrogels in the food industry, detailing their functionalities and contributions. Meanwhile, the intersection of 3D printing and food production has been exemplified by Park et al. [124] through innovative work involving callus-based 3D printing, particularly using carrot tissues. This approach showcases the potential for revolutionizing food manufacturing processes. Another promising avenue is the application of hydrogels based on ozonated cassava starch, as investigated by Maniglia et al. [125]. Their research focused on the impact of ozone processing and gelatinization conditions, particularly in the context of enhancing 3Dprinting applications. Additionally, the exploration of pectin hydrogels in food applications has been comprehensively reviewed by Ishwarya and Nisha [126], shedding light on the recent advances and future prospects in leveraging pectin hydrogels for diverse food-related functionalities. 

#### 4.2.2. Cosmetic Industry

In the cosmetics industry, hydrogel 3D printing offers the potential to develop novel skincare products with enhanced absorption and release properties. Customizable shapes and designs enable the creation of cosmetics that adhere better to the skin. Several cosmetic industries are using hydrogel 3D printing to create intricate face masks that fit snugly on the face, enhancing the delivery of skincare ingredients and improving the overall application experience. Finny et al. [127] delved into the realm of sun protection with their work on 3D-printed hydrogel-based sensors designed for quantifying UV exposure. This innovation holds immense potential for personalized UV monitoring. Diogo et al. [128] explored the engineering of hard tissues using cell-laden biomimetically mineralized shark-skin-collagen-based 3D-printed hydrogels, offering a promising avenue for developing advanced biomedical materials. The intersection of 3D printing and dermatology is evident in the work of de Oliveira et al. [129], who focused on 3D-printed products for topical skin applications, ranging from personalized dressings to drug delivery systems. Othman et al. [130] contributed to the field of cosmetic science by developing easy-to-use and cost-effective sensors for assessing the quality and traceability of cosmetic antioxidants, addressing a critical need in the cosmetic industry. 

#### 4.2.3. Electronics

Hydrogel 3D printing is also being explored for applications in electronics due to its unique combination of properties, including high water content and the potential for conductive or responsive behavior. This technology can enable the creation of soft, flexible electronics. Hydrogel-based sensors with 3D-printed structures can be used in wearable devices to monitor physiological parameters, such as heart rate and hydration levels. Guo et al. [131] introduced the 3D printing of electrically conductive and degradable hydrogel for epidermal strain sensors, which hold promise in wearable electronics.

Liu et al. [132] introduced a cost-effective, specialized 3D printing technology, direct ink writing (DIW), for creating hydrogel sensors enhanced with two-dimensional transition metal carbides (MXenes). These sensors exhibit exceptional strain and temperature sensing capabilities for shape memory solar array hinges. In their approach, Figure 9a illustrates the MXenes preparation process, which involves a fluoride-based salt etching system to minimize the risks associated with concentrated hydrogen fluoride (HF) dilution. Notably, this process uses LiF crystals dissolved in a hydrochloric acid (HCl) aqueous solution to gradually form HF with lower concentration under low-temperature conditions. This mixture is then blended with Ti_3_AlC_2_ powders to create a black suspension, ensuring sufficient etching without excessive oxidation.

Subsequent steps involve washing, intercalation, sonication, and argon protection to obtain delaminated Ti_3_C_2_T_x_ flakes. Subsequently, extended centrifugation was employed to separate the delaminated Ti_3_C_2_T_x_ flakes from the multi-layered aggregates. Given the presence of electronegative groups on Ti_3_C_2_T_x_, polymer hydrogel networks were established, as depicted in Figure 9b. The typical procedure involved the even dispersion of lyophilized Ti_3_C_2_T_x_ flakes, followed by the addition of glycerol and polymers. The resulting black mixture was heated to facilitate the dissolution of PVA under an argon (Ar) atmosphere, preventing unnecessary oxidation of Ti_3_C_2_T_x_ due to high temperatures. The resulting liquid precursor was then injected into a syringe and utilized in the DIW printing process. The SEM images provided in Figure 9c,d depict the Ti_3_C_2_T_x_ flakes displaying a sleek 2D nanosheet morphology, with lateral sizes ranging from 0.4 to 1 μm. As shown in the atomic force microscope (AFM) in Figure 9e, these Ti_3_C_2_T_x_ flakes exhibit a smooth surface with an average thickness of 1.6 nm, consistent with the thickness of a Ti_3_C_2_T_x_ bilayer structure. To examine the intricate internal microstructure of the hydrogel networks, the printed hydrogel underwent a process of lyophilization following solvent replacement. As illustrated in Figure 9f,g, the unaltered printed hydrogel showcases a porous network structure characterized by random orientation and a hierarchical distribution of pore sizes. The cross-sectional morphology of the printed hydrogel, along with its corresponding energy-dispersive X-ray spectroscopy (EDS) elemental mapping, is displayed in Figure 9h, confirming the uniform distribution of elements C, O, and N. Analyzing the EDS element distribution, it becomes evident that the relative content of N elements is lower than that of C and O. This indicates that the hard domains, comprised of nitrogen-containing groups within the polyurethane (PU) network, are less abundant than the soft domains, which in turn impacts the mechanical strength of the hydrogel. Similarly, when 0.25 wt% Ti_3_C_2_T_x_ flakes are introduced into the precursor, the resulting printed hydrogel exhibits an unoriented porous framework with random pore size distribution, resembling that of the pristine hydrogel, as depicted in Figure 9i,j. Furthermore, EDS analysis of elements C, O, and Ti (Figure 9k) confirms the homogeneous distribution of Ti_3_C_2_Tx flakes within the hydrogel.

Yang et al. [133] summarized recent advances in the 3D printing of electrically conductive hydrogels for flexible electronics, demonstrating their significance in developing next-generation electronic devices. Zhang and collaborators [134] also provided a review on 3D-printing hydrogels for actuators, offering insights into the development of responsive materials for various applications. Chen et al. [135] explored the utilization of 3D-printed hydrogels in the development of soft thermo-responsive smart windows, showcasing their adaptability in architectural and environmental applications. Guo and colleagues [136] investigated the use of biomass-derived hybrid hydrogel evaporators for cost-effective solar water purification, providing sustainable solutions for clean water access.

## 5. Current Challenges and Future Trends in Hydrogel 3D Printing

### 5.1. Current Challenges

Inspite of the various versatile properties and applications of hydrogel 3D printing, several challenges associated with it persist, such as resolution limitations, post-printing processing, scalability, etc. Achieving high-resolution printing remains a challenge due to the complex rheological properties of hydrogels. The precise deposition of fine features is hindered by factors like nozzle clogging and the tendency of hydrogels to spread upon deposition. Also, many hydrogel 3D printing methods require post-processing steps, such as curing or crosslinking, to enhance mechanical stability, whichcan be time-consuming and may introduce variability in the final product. Scaling up hydrogel 3D printing to produce large quantities of intricate structures is also challenging. Maintaining consistent print quality, gelation kinetics, and resolution across a larger volume is a significant hurdle.

### 5.2. Future Trends and Advancements

Hydrogel 3D printing faces current challenges related to resolution, post-printing processing, and scalability. The future holds promising trends and advancements. As materials, techniques, and applications continue to evolve, hydrogel 3D printing is poised to reshape industries and contribute to breakthroughs in personalized medicine, tissue engineering, and various other fields. Future advancements will likely introduce innovative hydrogel formulations with improved printability, biocompatibility, and mechanical properties. Hybrid hydrogels combining natural and synthetic components might become more prevalent, allowing for tailored materials for specific applications. Researchers are likely to develop improved printing techniques that enhance resolution and speed. This might include advancements in nozzle design, improved droplet formation in inkjet-based systems, and refined light-based curing methods in SLA and DLP. Advances in multi-material 3D printing could enable the integration of hydrogels with other materials, such as polymers and ceramics, allowing for the creation of complex and functional hybrid structures.

Hydrogel 3D printing will likely explore incorporating bioactive agents directly into printed structures. This could lead to the creation of constructs with integrated therapeutic properties, reducing the need for post-printing functionalization. Research into intricate scaffold architectures will continue, focusing on enhancing tissue integration, nutrient transport, and cellular responses. This could involve the exploration of gradient structures and hierarchical designs. The field might see an expansion into personalized medicine, with the capability to tailor hydrogel formulations and structures for individual patients. Point-of-care 3D printing could become feasible for producing patient-specific implants and drug delivery systems. As hydrogel 3D printing matures, more efforts will be directed toward regulatory approval and clinical translation of hydrogel-based medical devices and implants. 

As technology continues to evolve, 4D printing technologies havealso been developed for hydrogel printing. Very recently, Wang and Guo [137] provided insights into recent advances in 4D-printing hydrogels designed for biological interfaces and delvedinto the innovative use of hydrogels that exhibit dynamic shape changes over time, showcasing the potential for applications in biological systems. Seo et al. have also introduced a hydrogel production platform that utilizes photo-cross-linkable and temperature-reversible chitosan polymer in combination with stereolithography 4D printing technology. This platform enables the creation of hydrogels with dynamic movement, opening up new possibilities in tissue engineering and regenerative medicine.

## 6. Conclusions

In the realm of 3D printing, the union of hydrogels and technology has fostered a dynamic landscape ripe with innovation and promise. This review has traversed the rich tapestry of hydrogel 3D printing, elucidating its profound impact on diverse domains and hinting at the limitless possibilities on the horizon. From their humble beginnings as unique, water-rich materials, hydrogels have evolved into versatile substrates capable of being sculpted into intricate structures with unparalleled precision. The convergence of hydrogels and 3D printing has unlocked doors to a myriad of applications spanning biomedical engineering, industrial production, and beyond. The biomedical sector, in particular, stands as a beacon of hope for hydrogel 3D printing’s transformative power. The creation of tissue-engineered constructs, regenerative implants, and precise drug delivery systems has revolutionized healthcare and holds promise for further breakthroughs. The ability to replicate the intricacies of human biology, coupled with the advent of personalized medicine, offers a glimpse into a future where hydrogel 3D printing plays a pivotal role in enhancing the quality of life. Yet, this review acknowledges that challenges persist. Resolution limitations, post-printing processing intricacies, and the quest for scalable manufacturing methods are reminders that there is still much terrain to conquer.

Looking forward, the future appears promising. The trajectory of hydrogel 3D printing is set to be marked by advanced material development, enhanced printing techniques, and novel applications that defy current boundaries. As we journey further into this frontier, we anticipate the emergence of innovative solutions, new industries shaped by hydrogel capabilities, and a profound impact on scientific and technological advancement.

## Figures and Tables

**Figure 1 gels-09-00960-f001:**
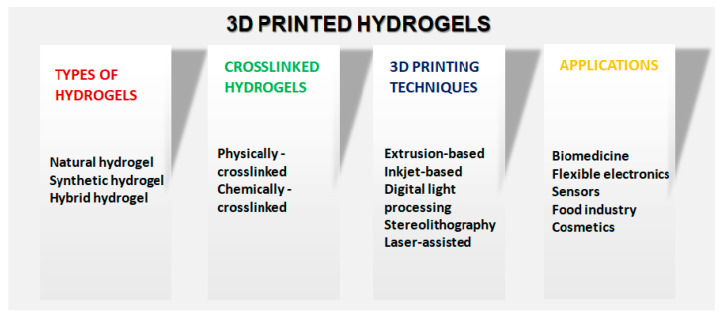
Schematic representation of the various types of hydrogels and their printing techniques that can be employed in several applications in various sectors, including biomedicine, flexible electronics, sensors, food industries, and cosmetics.

**Figure 2 gels-09-00960-f002:**
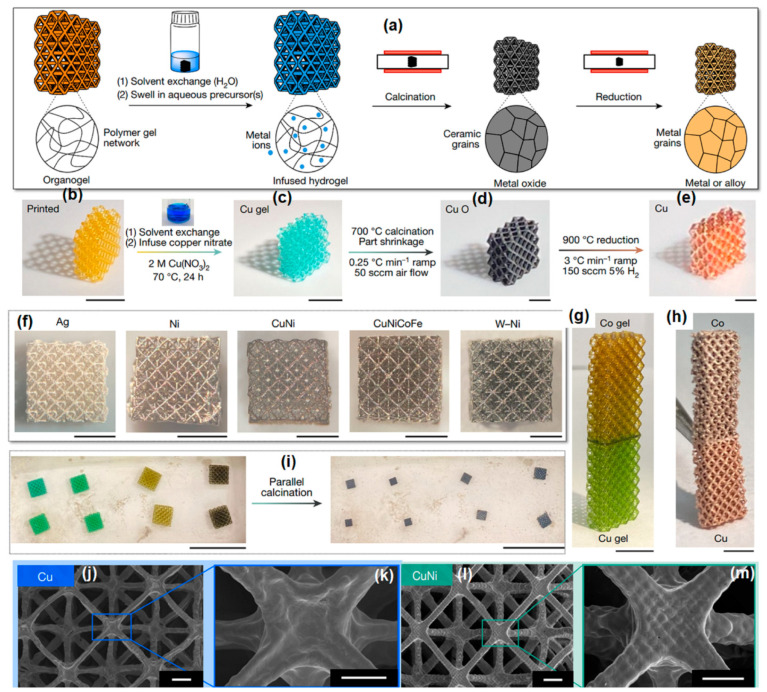
Hydrogel-infused additive manufacturing process and materials. (**a**) Illustration of the HIAM process. A 3D-printed organogel structure, composed of DMF/PEGda, undergoes a transformation into an infused hydrogel replica through the sequential steps of photoactive compound removal, solvent exchange, and infusion with a suitable aqueous precursor. Subsequent calcination in an oxygen environment leads to the formation of metal oxide structures, which are subsequently reduced to metals in forming gas. (**b**–**e**) Visual representations of the HIAM process for Cu metal: the printed organogel (**b**), the infused hydrogel (**c**), the calcined metal oxide (**d**), and the reduced metal (**e**). (**f**) Diverse metals and alloys fabricated using HIAM, including Ag, Ni, a binary CuNi alloy, a high-entropy CuNiCoFe alloy, and a refractory W–Ni alloy. (**g**) An octet lattice infused with Cu(NO_3_)_2_ on one end and Co(NO_3_)_2_ on the other. (**h**) After calcination and reduction, the Cu/Co gel transforms into a Cu/Co multi-material. (**i**) Simultaneous calcination of various infused gels. (**j**,**k**) and (**l**,**m**) SEM micrographs of Cu and CuNi octet lattices, depicting several unit cells from an overhead view (**j**,**l**) and a single junction node (**k**,**m**), respectively. (Scale bars: (**j**,**l**), 100 µm; (**k**,**m**), 50 µm). Reprinted from [25] under a Creative Commons Attribution License 4.0 (CC BY) “https://creativecommons.org/licenses/by/4.0/ (accessed on 1 November 2023)”.

**Figure 3 gels-09-00960-f003:**
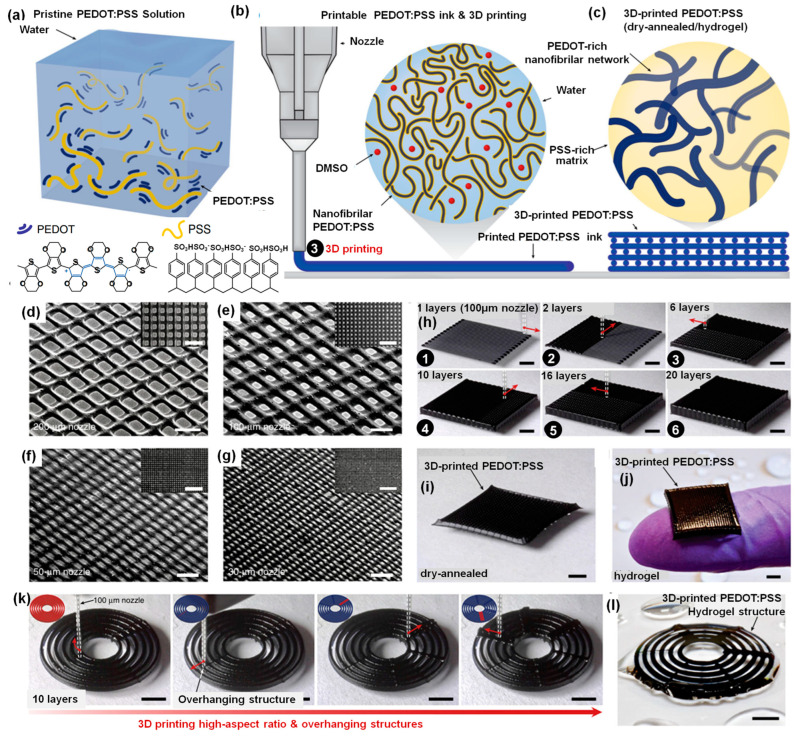
Development of a 3D-printable conductive polymer ink design. (**a**,**b**)The initial pristine PEDOT:PSS solution (**a**) can undergo a transformation into a 3D-printable conductive polymer ink (**b**) through lyophilization under cryogenic conditions followed by re-dispersion using an appropriate solvent. (**c**) 3D-printed conductive polymers can be transformed into pure PEDOT:PSS, whether in their dry state or hydrogel state, through a process involving dry annealing and subsequent expansion in a wet environment. (**d**–**g**) SEM micrographs of 3D-printed conducting polymer meshes. (**h**) Step-by-step progression of creating a 20-layered meshed structure using the conducting polymer ink. (**i**) 3D-printed conducting polymer mesh after undergoing a dry-annealing process, crucial for enhancing its conductivity. (**j**) 3D-printed conducting polymer mesh in its hydrogel state, highlighting its adaptability to different environmental conditions. (**k**) Sequential snapshots for 3D printing overhanging features across high aspect ratio structures using the conducting polymer ink, demonstrating its ability to maintain structural integrity during complex printing. (**l**) 3D-printed conducting polymer structure with overhanging features transformed into a hydrogel state. Reprinted from [28] under a Creative Commons Attribution License 4.0 (CC BY) “https://creativecommons.org/licenses/by/4.0/ (accessed on 1 November 2023)”.

**Figure 4 gels-09-00960-f004:**
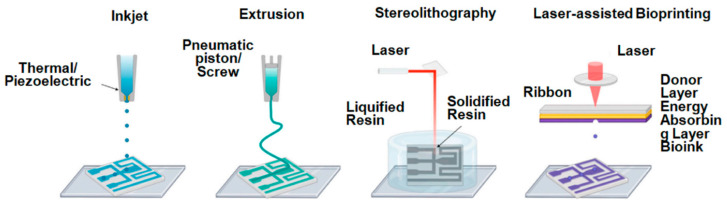
Illustration depicting diverse 3D printing techniques. Reprinted from [52] under a Creative Commons Attribution License 4.0 (CC BY) “https://creativecommons.org/licenses/by/4.0/ (accessed on 1 November 2023)”.

**Figure 5 gels-09-00960-f005:**
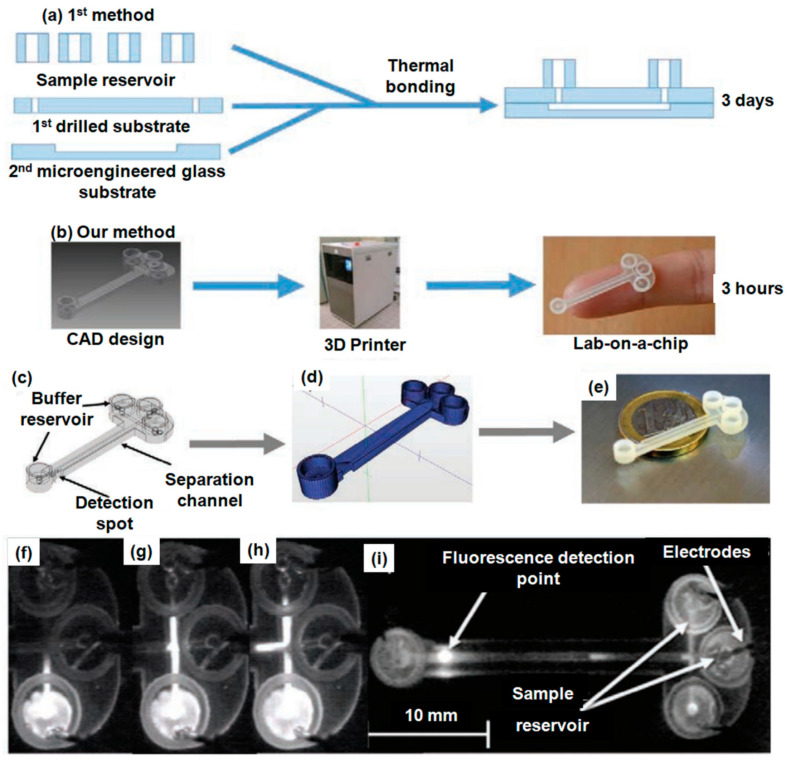
Sequential steps in the development and application of a microfluidic chip for gel electrophoresis. (**a**) Schematic presentation of the microfluidics chip manufacturing process through glass microengineering. (**b**) CAD design and the subsequent inkjet 3D printing of the lab-on-a-chip. (**c**) The original CAD design. (**d**) The exportation process leading to the creation of a geometric file. (**e**) The final structure offering a size comparison to a coin. In the investigation of DNA samples, (**f**–**h**) delineate the process of introducing an electrophoretic gel into the chip’s injection microchannel and subsequently injecting it into the separation microchannel. (**i**) Depicts the optical detection method involving fluorescence modulation via a laser light and a CCD mini camera. Reprinted from [66] under a Creative Commons Attribution License 4.0 (CC BY) “https://creativecommons.org/licenses/by/4.0/ (accessed on 1 November 2023)”.

**Figure 6 gels-09-00960-f006:**
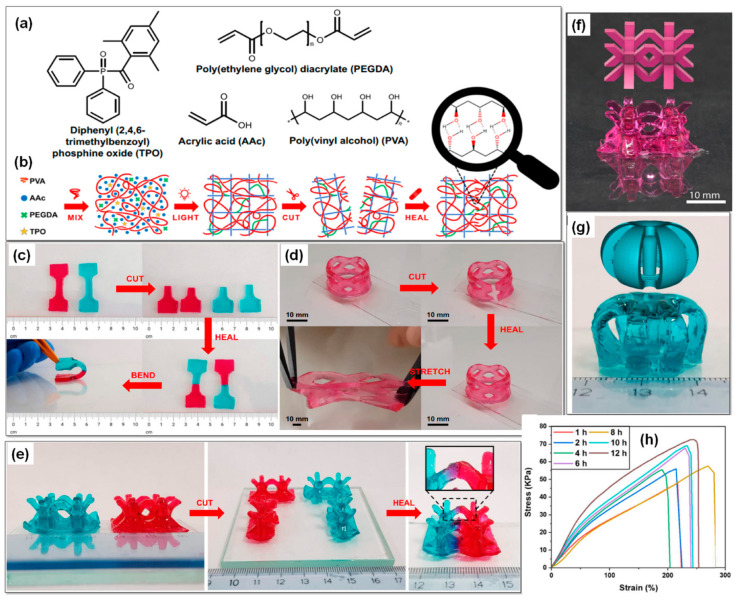
Composition formulation and network formation. (**a**) Molecular structure of initiator, monomer, cross-linker, and mending agent within the photocurable resin. (**b**) Schematic depiction of semi-IPN and the healing process. Illustrated recovery of cut and rejoined objects printed with an AAc/PVA ratio of 0.8. (**c**) Tensile test samples featuring methyl red sodium salt and brilliant green dye. These specimens exhibited immediate bending deformation upon rejoining. (**d**) A cylindrical structure with holes printed using methyl red sodium salt dye. The reconnected sample demonstrated stretching deformation endurance after a 2h healing process. (**e**) Lattice-like structures with a body-centered cubic pattern, printed with methyl red sodium salt and brilliant green dye. Noticeable dye diffusion at the interface is visible after 12 h (inset) due to the dye concentration gradient. (**f**,**g**) 3D-printed samples with the PVA 0.8 formulation: (**f**) body-centered cubic lattice-like structure printed with methyl red sodium salt dye. (**g**) Axisymmetric structure featuring a central pillar, printed with brilliant green dye. (**h**) Stress–strain curves of self-healed hydrogels over rising healing periods. Reprinted from [89] under a Creative Commons Attribution License 4.0 (CC BY) “https://creativecommons.org/licenses/by/4.0/ (accessed on 1 November 2023)”.

**Figure 7 gels-09-00960-f007:**
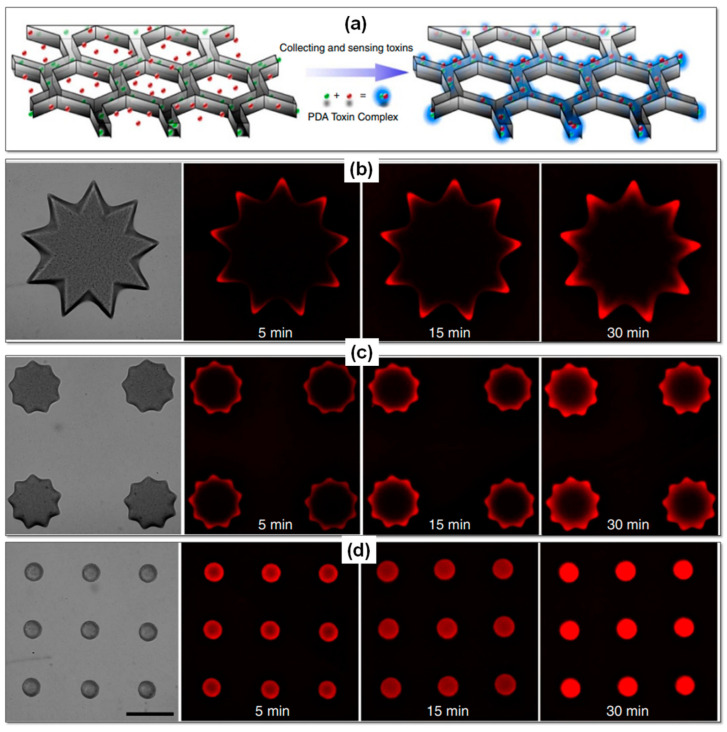
(**a**) PDA nanoparticles (green) embedded within a PEGDA hydrogel matrix (grey) to create a liver-mimetic 3D structure through 3D printing. Nanoparticles possess the capability to attract, capture, and sense toxins (red), while the modified liver lobule structure within the 3D matrix allows for efficient toxin entrapment. (**b**–**d**) Toxin capture within 3D-printed hydrogel nanocomposites featuring various surface patterns. Three distinct types of 3D structural posts, sharing a similar flower-like form and length but differing diameters, undergo incubation at 37 °C with a melittin solution (50 mg/mL). (**b**) Demonstrates a post with a large diameter. (**c**) Represents the medium-diameter post. (**d**) Features the narrow-diameter post. The red fluorescence highlights the interactions between PDA and melittin, marking the toxic region. A scale bar of 200 mm is provided for reference. Reprinted from [102] under a Creative Commons Attribution License 4.0 (CC BY) “https://creativecommons.org/licenses/by/4.0/ (accessed on 1 November 2023)”.

**Figure 8 gels-09-00960-f008:**
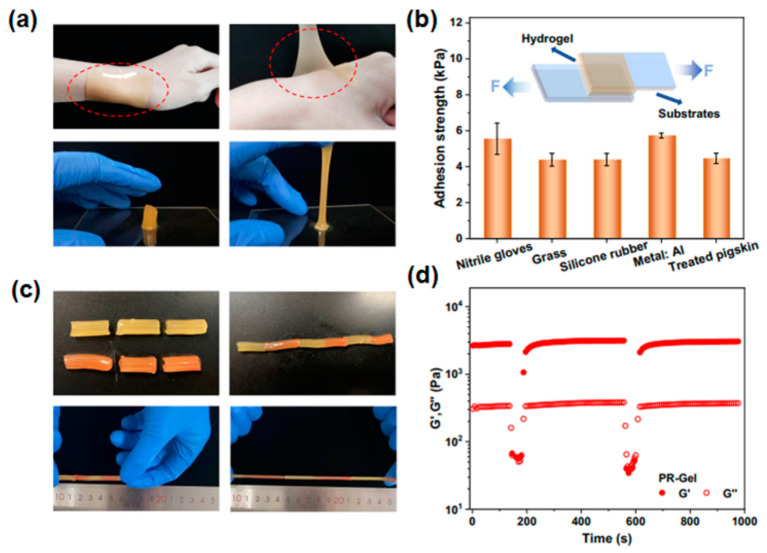
Shear adhesive strength and self-healing: (**a**) photograph of PR-Gel (40 *w*/*v*%) stuck onto wrist joints and a glass surface. (**b**) Represents the shear adhesive strength of PR-Gel (40 *w*/*v*%) on various substrates, including a nitrile glove, glass plate, silicone rubber, aluminum, and pigskin. The inset offers a schematic illustration of the lap-shear test conducted at 25 °C. (**c**) Self-healing behavior of the PR-Gel (40 *w*/*v*%) at 25 °C, with the visualization aided by rhodamine B. (**d**) Storage modulus (G′) and loss modulus (G″) of the PR-Gel (40 *w*/*v*%) under alternating strains, oscillating between small (1%) and large strain (500%) at 25 °C. Reprinted from [103] under a Creative Commons Attribution License 4.0 (CC BY) “https://creativecommons.org/licenses/by/4.0/ (accessed on 1 November 2023)”.

**Figure 9 gels-09-00960-f009:**
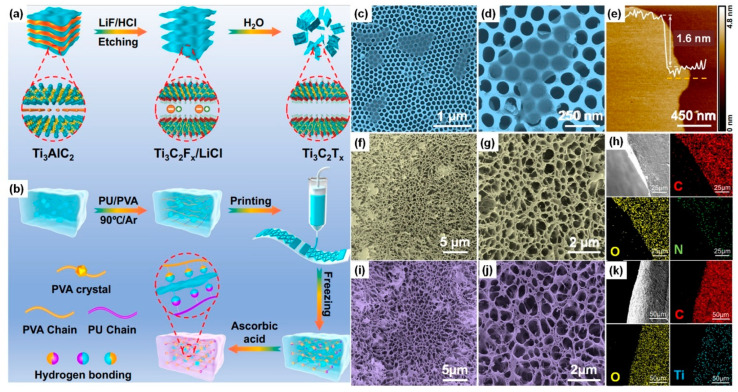
Synthesis of MXenes and hydrogel: (**a**) pictorial representation of the synthesis process of Ti3C2Tx flakes. (**b**) Demonstration of DIW printing of Mxenes-attached polyurethane/polyvinyl alcohol (PU/PVA) hydrogel. Morphology and structure of materials: (**c**,**d**) SEM morphology and (**e**) AFM image of Ti3C2Tx flakes. (**f**,**g**) Microstructure of pristine lyophilized gel and (**h**) corresponding elements distribution. (**i**,**j**) Microstructure of Ti3C2Tx-loaded lyophilized gel and (**k**) depicts element distribution within this structure. Reprinted from [132] under a Creative Commons Attribution License 4.0 (CC BY) “https://creativecommons.org/licenses/by/4.0/ (accessed on 1 November 2023)”.

**Table 1 gels-09-00960-t001:** Comparative summary of the properties of natural, synthetic, and hybrid hydrogels in terms of preparation, printability, biocompatibility, and mechanical properties.

Property	Natural Hydrogels	Synthetic Hydrogels	Hybrid Hydrogels
**Preparation**	- Derived from biological sources like alginate, collagen, or agarose. - Typically requires minimal chemical modification.	- Chemically synthesized with precise control over composition. - Tailored to specific requirements through chemical synthesis.	- Combination of natural and synthetic components. - Utilizes the advantages of both natural and synthetic components.
**Printability**	- Varies depending on source material. - May require additional modifications for 3D printing.	- Generally good printability due to controlled composition. - Compatible with various 3D printing techniques.	- Printability depends on the combination and ratio of components. - Printability can be customized for specific applications.
**Biocompatibility**	- Generally biocompatible due to natural origin. - Often suitable for cell encapsulation and tissue engineering.	- Biocompatibility can vary based on the polymer used. - Careful selection of synthetic components can enhance biocompatibility.	- Biocompatibility influenced by natural components. - Potential for improved biocompatibility through hybridization.
**Mechanical Properties**	- Mechanical properties can vary widely depending on the source. - May have lower mechanical strength compared to synthetics.	- Can be precisely tuned for specific applications. - Offers a wide range of mechanical properties (soft to stiff).	- Mechanical properties can be tailored to desired levels. - Balances natural properties with synthetic enhancements.

**Table 2 gels-09-00960-t002:** Description of the various properties of hydrogels along with their corresponding applications.

Property	Description	Applications
Water Absorption and Retention	Ability to absorb and retain a significant amount of water or biological fluids.	Wound dressings, contact lenses, diapers, and drug delivery systems.
Biocompatibility	Compatibility with living tissues, making hydrogels suitable for medical and biological applications.	Tissue engineering, drug delivery, wound healing, and surgical implants.
Tunable Mechanical Properties	Adjustability of mechanical characteristics like elasticity and stiffness for specific applications.	Cartilage replacements, soft tissue engineering, and drug delivery matrices.
Swelling Behavior	Controllable ability to swell in response to factors such as pH, temperature, or ionic strength.	Controlled drug release, biosensors, and wound dressings.
Permeability	Ability to allow the passage of certain substances while restricting others.	Controlled drug delivery, filtration membranes, and biosensors.
Responsive to Stimuli	Capability to undergo changes in volume or structure in response to external stimuli (e.g., temperature, pH, and light).	Smart drug delivery, biosensing, and controlled release systems.
Adhesive Properties	Ability to adhere to biological tissues, useful for applications like wound dressings and tissue adhesives.	Surgical adhesives, wound closures, and tissue engineering.
Versatility	Wide range of options in terms of composition and structure, allowing customization for various applications.	Tissue engineering scaffolds, wound dressings, and drug delivery systems.
Electrical Conductivity	Capability to conduct electricity, often achieved by incorporating conductive polymers or nanoparticles.	Biosensors, flexible electronics, and neural interfaces.

**Table 3 gels-09-00960-t003:** Influence and definition of various key parameters and properties under rheology, gelation kinetics, and print fidelity on 3D printing processes.

Parameter/Properties	Definition	Effect on 3D Printing
**RHEOLOGICAL PROPERTIES**		
Viscosity	Viscosity measures a fluid’s resistance to flow.	Affects the ease of handling and deposition.
Shear-Thinning Behavior	Shear-thinning is the property where viscosity decreases under shear stress.	Facilitates smooth flow during printing; hydrogel becomes less viscous when subjected to shear stress, allowing for easy extrusion.
Thixotropy	Property where a material becomes less viscous over time under constant shear stress and recovers its viscosity when the stress is removed.	Allows the hydrogel to recover its original viscosity between printing layers, preventing spreading and maintaining structural integrity.
Viscoelasticity	Viscoelastic materials exhibit both viscous (flow) and elastic (deformation recovery) properties.	Affects the material’s response to stress and strain during printing and ensures the printed structure retains its shape after deposition.
Gelling Mechanisms	Refers to the process by which a liquid transforms into a gel or solid.	Determines speed and control of gelation, influencing the overall printing process and final construct.
Extrudability	The ease with which a material can be extruded or forced through a nozzle.	Affects the precision and control of material deposition during 3D printing.
Shape Retention	The ability of the material to maintain its intended shape after deposition.	Critical for achieving accurate and consistent layer-by-layer printing, ensuring the final structure matches the design.
**GELATION KINETICS**		
Gelation Time	The time taken for a liquid to transition into a gel.	Affects the overall printing speed and duration of the 3D printing process.
Crosslinking Density	The concentration of crosslinks formed between polymer chains during gelation.	Affects the mechanical strength and stability of the resulting hydrogel.
Temperature	The degree of heat applied during the gelation process.	Affects the rate of chemical reactions or physical processes leading to gel formation.
Concentration of Crosslinking Agents	The amount of crosslinking agents, such as chemical initiators, present in the hydrogel formulation.	Higher concentrations typically result in faster gelation but may impact other material properties.
pH Level	The acidity or alkalinity of the hydrogel formulation.	Affects the ionization of functional groups and, hence, the gelation process.
Polymer Concentration	The concentration of polymer molecules in the hydrogel formulation.	Higher concentrations can lead to denser networks and affect the gelation time and mechanical properties.
Solvent Composition	The type and ratio of solvents used in the hydrogel formulation.	Solvent properties can impact the rate of gelation and the resulting structure of the hydrogel.
Initiator Concentration (Photochemical Gelation)	The concentration of photoinitiators in the hydrogel formulation.	Critical for photochemical gelation processes, where light triggers crosslink formation.
Stirring Rate (for Physical Gelation)	The speed at which the hydrogel components are mixed.	Affects the distribution of components and can influence the gelation time in physically gelled hydrogels.
Presence of Catalysts	Chemical substances that accelerate gelation reactions.	Help in enhancing the speed and efficiency of gelation processes.
**PRINT FIDELITY**		
Layer Resolution	The thickness of each layer deposited during printing.	Finer resolutions lead to smoother surfaces and improved details.
Extruder Calibration	Adjusting the extruder to ensure accurate material deposition.	Prevents under- or over-extrusion, enhancing print accuracy.
Bed Leveling	Ensuring the print bed is perfectly level.	Prevents uneven layer heights, promoting uniform adhesion.
Print Speed	The speed at which the printer deposits material.	Optimizing print speed balances accuracy with efficiency; too fast can lead to errors.
Temperature Control	Maintaining consistent temperatures for the printer and printing material.	Fluctuations can affect material flow and layer adhesion.
Material Quality	The quality and consistency of the printing material.	Inconsistent materials may lead to variations in print quality.
Print Bed Adhesion	Ensuring the first layer adheres well to the print bed.	Proper adhesion prevents warping and helps maintain accurate layer alignment.
Support Structures	Temporary structures to support overhanging features.	Well-designed supports prevent deformations and maintain accuracy.
Cooling Systems	Fans or other cooling mechanisms to solidify layers quickly.	Proper cooling prevents overheating and improves feature definition.
Printer Rigidity	The stability and rigidity of the printer frame.	A stable frame reduces vibrations and ensures precise movements.
Print Orientation	The angle and direction in which the object is printed.	Optimal orientation minimizes overhangs and supports, improving print fidelity.
Filament Diameter	The diameter of the printing filament.	Accurate filament diameter ensures consistent material flow.
Environmental Conditions	Factors like temperature and humidity in the printing environment.	Extreme conditions can affect material properties and printing outcomes.
Print Design	The complexity and geometry of the printed object.	Complex designs may require specific settings for accurate printing.

**Table 4 gels-09-00960-t004:** Extrusion-based 3D-printed hydrogel varieties and their diverse applications.

Hydrogel Used	Applications/Remarks	Reference
Laponite^®^ incorporated oxidized alginate–gelatin composite hydrogels	Integration of natural hydrogels into advanced manufacturing processes	Cai et al., (2021) [23]
Cellulose derivative	Biodegradable support material	Cheng, Y. et al., (2020) [53]
Gelatin-oxidized nanocellulose hydrogels	Bioprinting applications for tissue engineering	Zhou, S. et al., (2022) [54]
Gelatine methacrylate/Laponite nanocomposite hydrogel	High-concentration nanoclay for bone tissue regeneration	Dong, L. et al., (2021) [55]
Pre-crosslinked alginate	Advanced printable hydrogels	Falcone, G. et al., (2022) [56]
Photoresponsive polypeptide hydrogels	Stable constructs composed of photoresponsive polypeptide hydrogels	Murphy, R. D. et al., (2019) [57]
Cellulose hydrogel	Cellulose hydrogel skeleton usingextrusion 3D printing of solution	Hu, X. et al., (2020) [58]
Multicomponent hydrogel-based bioinks	Bioprinting for various applications using multicomponent hydrogel-based bioinks	Cui, X. et al., (2020) [59]
Chitosan hydrogels	Simulations of extrusion 3D printing of chitosan hydrogels	Ramezani, H. et al., (2022) [60]

**Table 5 gels-09-00960-t005:** Inkjet-based 3D-printed hydrogels and their diverse applications.

Hydrogel Used	3D Printing Technology	Applications/Remarks	Reference
Itraconazole nanocrystals on hydrogel	Inkjet Printing	Ophthalmic drug delivery	Tetyczka, C. et al., (2022) [68]
Hydrogel-based bioinks	Thermal inkjet bioprinting	Printability improvement via saponification and heat treatment processes	Suntornnond, R. et al., (2022) [61]
Hyaluronic acid hydrogel	Photolithography and light-cured inkjet printing methods	Comparing different methods for creating hyaluronic acid hydrogel micropatterns	Chen, F. et al., (2022) [75]
Hydrogel	3D Inkjet Printing	Cell-laden structures Complex tissue engineering	Negro et al., (2018) [67]
Hydrogel	Microreactive Inkjet Printing	Free-standing 3D microstructures Microscale devices	Teo et al., (2019) [69]
Star block copolymer hydrogels	3D Inkjet Printing	Cross-linked with metallic ions Structural materials	Nakagawa et al., (2017) [70]
Alginate/gelatin hydrogel	Inkjet Printing	Mechanical and biological properties Tissue engineering	Jiao et al., (2021) [71]
Multilayered hydrogel	Inkjet–Spray Hybrid Printing	Multilayered hydrogel structures Multifunctional constructs	Yoon et al., (2018) [72]
Multilayered hydrogel	Ion Inkjet Printing	Surface patterning for shape deformations Programmable structures	Peng et al., (2017) [73]
Poly-ɛ-lysine/gellan gum hydrogels	Reactive Inkjet Printing	Corneal constructs Ophthalmic applications	Duffy et al., (2021) [74]

**Table 6 gels-09-00960-t006:** Stereolithography-based 3D-printed hydrogels and their diverse applications.

Hydrogel Used	Applications/Remarks	Reference
Nanocellulose/PEGDA	3D cell culture, tissue engineering	Tang, A. et al., (2019) [29]
Nanocomposite hydrogels	Creating hydrogel structures with vascular networks	Kalossaka, L.M. et al., (2021) [76]
Ascorbic acid-loaded hydrogels	Controlled drug release from 3D-printed hydrogels	Karakurt, I. et al., (2020) [77]
PEGDMA hydrogels	Investigating the impact of 3D printing on hydrogel properties	Burke, G. et al., (2020) [78]
Hydrogel structures	Low-cost 3D printing of hydrogel structures	Magalhães, L.S.S. et al., (2020) [79]
PEGDA hydrogels	Studying the impact of photopolymerization in micro-stereolithography	Alketbi, A.S. et al., (2021) [80]
Hydrogels with tunability	Developing hydrogels with mechanical tunability and self-welding properties	Sun, Z. et al., (2022) [81]

**Table 7 gels-09-00960-t007:** DLP-based 3D-printed hydrogels and their diverse applications.

Hydrogel Used	Applications/Remarks	Reference
Hydrogels	Creating hydrogels with hierarchical structures	Sun, Z. et al., (2023) [84]
Supramolecular hydrogels	Tough supramolecular hydrogels with sophisticated architectures as impact-absorption elements	Dong, M. et al., (2022) [85]
Cellulose hydrogel	Cellulose hydrogel for strain sensing	Guo, Z. et al., (2023) [86]
Cellulose-based hydrogels	3D printing of fully cellulose-based hydrogels usingDLP	Cafiso, D. et al., (2022) [87]
Antimicrobial hydrogel	Developing an antimicrobial hydrogel using sustainable resin and hybrid nanospheres	Wang, L. et al., (2022) [88]
Antheraea pernyi silk fibroin	Using silk fibroin bioinks for DLP 3D printing	Zhang, X. et al., (2023) [90]
Double-network hydrogels	3D printing of high-toughness double network hydrogels	Xiang, Z. et al., (2022) [91]
PEDOT-based photopolymerizable inks	DLP 3D printing of PEDOT-based photopolymerizable inks for biosensing	Lopez-Larrea, N. et al., (2022) [92]

**Table 8 gels-09-00960-t008:** Comparative summary of various aspects of 3D printing technologies in terms of principle, resolution, material compatibility, speed, post-processing, advantages, and limitations.

	Extrusion-Based	Inkjet-Based	SLA	DLP
**Principle**	Layer-by-layer extrusion	Droplet deposition	Photopolymerization	Photopolymerization
**Resolution**	Moderate	High	Very High	Very High
**Material Compatibility**	Wide range of hydrogel materials	Limited to specific hydrogel formulations	Limited choice due to compatibility	Limited choice due to compatibility
**Speed**	Moderate	Moderate to Fast	Moderate to Fast	Very Fast
**Post-Processing**	Often minimal	May require curing	Minimal	Minimal
**Advantages**	- Versatile for various hydrogels. - Cost-effective and simple. - Well-suited for large-scale objects. - Handles high-viscosity inks well.	- Precise control. - Multi-material printing capabilities. - Suitable for biomedicine and research. - Excellent for fine-detail structures.	- Exceptional resolution and surface finish. - Complex and intricate geometries possible. - Fast printing speed for high-detail objects.	- Fast printing speed, suitable for large-scale production. - Capable of intricate geometries with precision.
**Limitations**	- Viscosity can lead to nozzle clogging. - Challenges with intricate structures. - Limited in applications requiring high precision.	- Prone to nozzle clogging, especially with viscous inks. - May require crosslinking. - Limited in creating large-scale objects.	- UV light exposure may affect cell viability. - Specialized equipment with high upfront cost. - -Potential resin waste if not fully used.	- UV light exposure may affect cell viability. - Requires specialized equipment and UV light source. - Potential resin waste if not fully used.

## Data Availability

Data sharing not applicable.

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
