# Peer review of "3D-Printed Hydrogel for Diverse Applications: A Review"

_gels, 2023, doi:10.3390/gels9120960_

Round 1
Reviewer 1 Report
Comments and Suggestions for Authors
This review presents the progress of 3D printing hydrogel for diverse applications, which is a meaningful work. As the whole manuscript was in bad organizing and writing, the review does not recommend publication at this stage.
(1) In the title, “Hydrogel 3D printing” should be “3D printing hydrogel”.
(2) Section “2.1. Classification of hydrogels” is very confused, which should further divided into different categories, such as 2.1.1 Natural Hydrogels, 2.1.2 Synthetic Hydrogels, 2.1.3 Hybrid Hydrogels, or other classification method.
(3) Section 2.2, the different properties and applications of hydrogels was present in a bad manner, which should present in tables respectively.
(4) In section 2.3, “2.2.1, 2.2.2” should be “2.3.1, 2.3.2”.
(5) Section 2.3, the formulations designing is not specific, which has no guidance. The formulations designing should present according to the processing parameters or properties.
(6) Section 4.1 and 4.2 should further divided into different categories as mentioned in the section of Abstract “It emphasizes the integration of hydrogel 3D printing in biomedical engineering, showcasing its role in 4.1.1 tissue engineering, 4.1.2 regenerative medicine, and 4.1.3 drug delivery. Beyond healthcare, it examines applications in industries like 4.2.1 food, 4.2.2 cosmetics, and 4.2.3 electronics. ”
(7) Line 872, “Table 3 summarizes the various 3D printed hydrogels employed for several applications. ” should deleted.
(8) Table 3 should present as the format of Table 1 and 2.
(9) 3D printed hydrogels present in Table 3 should divided into different tables according to the categories (4.1.1 tissue engineering, 4.1.2 regenerative medicine, 4.1.3 drug delivery and 4.2.1 food, 4.2.2 cosmetics, 4.2.3 electronics), and present at corresponding section.
(10) English language needs editing.
Comments on the Quality of English LanguageExtensive editing of English language required
Author Response
Review#01
This review presents the progress of 3D printing hydrogel for diverse applications, which is a meaningful work. As the whole manuscript was in bad organizing and writing, the review does not recommend publication at this stage.
Comment 1: In the title, “Hydrogel 3D printing” should be “3D printing hydrogel”.
Reply to the comment: Title has been now corrected in the revised manuscript as “3D Printed Hydrogel for Diverse Applications: A review”. (Page 1; title of the review article)
Comment 2: Section “2.1. Classification of hydrogels” is very confused, which should further divided into different categories, such as 2.1.1 Natural Hydrogels, 2.1.2 Synthetic Hydrogels, 2.1.3 Hybrid Hydrogels, or other classification method.
Reply to the comment: Classification of hydrogels has been now further divided into subsections such as 2.1.1 Natural Hydrogels, 2.1.2 Synthetic Hydrogels, 2.1.3 Hybrid Hydrogels, 2.1.4. Crosslinked hydrogels (Subsections of section 2.1 on pages 3-9)
Comment 3: Section 2.2, the different properties and applications of hydrogels was present in a bad manner, which should present in tables respectively.
Reply to the comment: A brief summary of the various properties of hydrogels along with their corresponding applications have been now presented in a tabular form in the revised manuscript. (Page 10; table-2)
Comment 4: In section 2.3, “2.2.1, 2.2.2” should be “2.3.1, 2.3.2”.
Reply to the comment: The sections/subsections have been carefully checked throughout the revised manuscript.
Comment 5: Section 2.3, the formulations designing is not specific, which has no guidance. The formulations designing should present according to the processing parameters or properties.
Reply to the comment: Formulations designing has been revised and is now presented specifically according to the processing parameters or properties. A table has also been included. (Section 2.3 on Page-10-13; Table 3 on page 12-13)
Comment 6: Section 4.1 and 4.2 should further divided into different categories as mentioned in the section of Abstract “It emphasizes the integration of hydrogel 3D printing in biomedical engineering, showcasing its role in 4.1.1 tissue engineering, 4.1.2 regenerative medicine, and 4.1.3 drug delivery. Beyond healthcare, it examines applications in industries like 4.2.1 food, 4.2.2 cosmetics, and 4.2.3 electronics. ”
Reply to the comment: Section 4.1 and 4.2 has been further divided into different categories in the revised manuscript as 4.1.1 tissue engineering (Page 23) 4.1.2. regenerative medicine (Page 25), and 4.1.2 drug delivery (Page 27), 4.2.1 food industry (Page 27), 4.2.2 cosmetic industry (Page 28), and 4.2.3 electronics (Page 28).
Comment 7: Line 872, “Table 3 summarizes the various 3D printed hydrogels employed for several applications. ” should deleted.
Reply to the comment: Line 872, “Table 3 summarizes the various 3D printed hydrogels employed for several applications. ” has been now deleted from the revised manuscript.
Comment 8: Table 3 should present as the format of Table 1 and 2. 3D printed hydrogels present in Table 3 should divided into different tables according to the categories (4.1.1 tissue engineering, 4.1.2 regenerative medicine, 4.1.3 drug delivery and 4.2.1 food, 4.2.2 cosmetics, 4.2.3 electronics), and present at corresponding section.
Reply to the comment: Thank you very much for your suggestion. Table 3 was originally designed to focus on the various 3D printing technologies employed for hydrogel. Accordingly, the table has been now divided into different tables according to 3D printing technology and presented at corresponding sections as described below-
Table 4. Extrusion based 3D printed hydrogel varieties and their diverse applications. (Page 15)
Table 5. Inkjet based 3D printed hydrogels and their diverse applications. (Page 18)
Table 6. Stereolithography-based 3D printed hydrogels and their diverse applications. (Page 19)
Table 7. DLP-based 3D printed hydrogels and their diverse applications. (Page 22)
All the tables have been now presented in the same format (as Table 1) in the revised manuscript.
Comment 9: English language needs editing.
Reply to the comment: We have carefully revised the manuscript and language has been corrected throughout the revised manuscript.
Reviewer 2 Report
Comments and Suggestions for Authors
Authors introduce a Review about Hydrogels in 3D Printing. The draft is clear, linear. The scientific value in all the Review drafts is by definition not a priority, may be for this type of draft a different evaluation scheme should be proposed in the future. The draft could be helpful as a starting point with these materials even if some more detailed and wide analysis of this technology would have been welcome in a Review draft.
Author Response
Review#02
Authors introduce a Review about Hydrogels in 3D Printing. The draft is clear, linear. The scientific value in all the Review drafts is by definition not a priority, may be for this type of draft a different evaluation scheme should be proposed in the future. The draft could be helpful as a starting point with these materials even if some more detailed and wide analysis of this technology would have been welcome in a Review draft.
Reply to Reviewer#02: Thank you for your constructive comments. We have tried our best to revise the manuscript.
Reviewer 3 Report
Comments and Suggestions for Authors
The review by these Authors regards a novel and interesting topic and I think it will be useful to those researchers approaching these applications of additive manufacturing. Despite some mistakes that can be revised (such as the overlapping of Additive manufacturing with 3D printing in the introduction, please revise AD is more than only 3D printing), the manuscript is worthy of publication. In the attached report other suggestions.

Author Response
Review#03
The review by these Authors regards a novel and interesting topic and I think it will be useful to those researchers approaching these applications of additive manufacturing. Despite some mistakes that can be revised (such as the overlapping of Additive manufacturing with 3D printing in the introduction, please revise AD is more than only 3D printing), the manuscript is worthy of publication. In the attached report other suggestions.
Reply to Reviewer#03: Thank you very much for your constructive and fruitful comments. We have carefully revised the manuscript as per the attached comments and have also included the suggested references regarding the use of 3D printing in microfluidic application.
- L Saitta, E Cutuli, G Celano, C Tosto, G Stella, G Cicala, M Bucolo, A Regression Approach to Model Refractive Index Measurements of Novel 3D Printable Photocurable Resins for Micro-Optofluidic Applications, Polymers 15 (12), 26901, 2023
- C Marzano, F Arcadio, A Minardo, L Zeni, D Del Prete, G Cicala, L Saitta Towards V-shaped Plasmonic probes made by exploiting 3D printers and UV-cured optical adhesives for Medical applications, , ...2023 IEEE International Workshop on Metrology for Industry 4.0 & IoT, 2023
- G Stella, L Saitta, AE Ongaro, G Cicala, M Kersaudy-Kerhoas, M Bucolo,Advanced Technologies in the Fabrication of a Micro-Optical Light Splitter, Micro 3 (1), 338-352 (2023)
- L Saitta, F Arcadio, G Celano, N Cennamo, L Zeni, C Tosto, G Cicala, Design and manufacturing of a surface plasmon resonance sensor based on inkjet 3D printing for simultaneous measurements of refractive index and temperature, The International Journal of Advanced Manufacturing Technology 124 (7-8 …
Round 2
Reviewer 1 Report
Comments and Suggestions for Authors
The format of Figure 3 and 7 should revised
Comments on the Quality of English LanguageModerate editing of English language required
Reviewer 2 Report
Comments and Suggestions for Authors
Authors provided a significant enhancement of the original draft. The second version is wider, the number of rererences has been extended. The scenario fits in a best way to a Review draft